# Structural insights into E1 recognition and the ubiquitin-conjugating activity of the E2 enzyme Cdc34

Katelyn M. Williams[1], Shuo Qie[1], James H. Atkison[1], Sabrina Salazar-Arango[1], J. Alan Diehl [1] & Shaun K. Olsen [1]

Ubiquitin (Ub) signaling requires the sequential interactions and activities of three enzymes, E1, E2, and E3. Cdc34 is an E2 that plays a key role in regulating cell cycle progression and requires unique structural elements to function. The molecular basis by which Cdc34 engages its E1 and the structural mechanisms by which its unique C-terminal extension functions in Cdc34 activity are unknown. Here, we present crystal structures of Cdc34 alone and in complex with E1, and a Cdc34~Ub thioester mimetic that represents the product of Uba1-Cdc34 Ub transthiolation. These structures reveal conformational changes in Uba1 and Cdc34 and a unique binding mode that are required for transthiolation. The Cdc34~Ub structure reveals contacts between the Cdc34 C-terminal extension and Ub that stabilize Cdc34~Ub in a closed conformation and are critical for Ub discharge. Altogether, our structural, biochemical, and cell-based studies provide insights into the molecular mechanisms by which Cdc34 function in cells.

[1] Department of Biochemistry & Molecular Biology and Hollings Cancer Center, Medical University of South Carolina, Charleston, SC 29425, USA. Correspondence and requests for materials should be addressed to S.K.O. (email: olsensk@musc.edu)

Reversible post translational modification of proteins by ubiquitin (Ub) regulates nearly every aspect of eukaryotic biology. Cell cycle progression is among the best studied processes controlled by Ub signaling and began with the observation that Cdc34, an E2 ubiquitin-conjugating enzyme, is essential for cell cycle progression in yeast[1,2]. As an E2, Cdc34 is the central player in a cascade of enzymes that includes an E1 ubiquitin activating enzyme, Uba1, and a large family of Skp-Cullin-F-box (SCF) E3 ubiquitin ligases that function in tandem to assemble Lys48-linked Ub chains on proteins to target them for proteasomal degradation[3–5]. Specifically, Cdc34 is best known to regulate G1-S checkpoint progression by targeting cyclin-dependent kinase inhibitors such as Sic1/p27 for degradation[6–9]. More recently, Cdc34 has been implicated in human pathologies including elevated Cdc34 levels in several cancers[10–14] and Cdc34 regulation of oncoproteins[15,16]. Accordingly, expression of a dominant-negative Cdc34 mutant enhances anti-myeloma activity of Bortezomib[17] and cisplatin therapy inhibits Cdc34-mediated degradation of ATF5, a prosurvival transcription factor[18], generating interest in the SCF-Cdc34 pathway as a potential therapeutic target[19–21].

Uba1 is a multidomain enzyme that serves as the gatekeeper of the Ub conjugation cascade by activating Ub in a two-step process involving adenylation and thioester bond formation followed by transfer of Ub to E2s in a process termed E1-E2 thioester transfer or transthiolation[3,22,23]. Each Uba1 domain plays a distinct functional role with active and inactive adenylation domains (AAD and IAD) that adenylate the C-terminus of Ub[24–27]; a Cys domain (split into first and second catalytic cysteine half domains, SCCH and FCCH[28]) that harbors the catalytic cysteine for thioester bond formation with Ub; and a ubiquitin fold domain (UFD) that is involved in molecular recognition of E2s[25,29,30]. Cdc34 is one of tens of E2s that must function with Uba1 despite significant differences at their predicted UFD-interacting surfaces. For example, Cdc34 lacks a conserved basic motif known to be important for E1-E2 thioester transfer for other E2s[31]; has an N-terminal extension proximal to the major UFD-binding region of E2s; and harbors amino acids with very different physiochemical properties at predicted E1-binding positions. Thus, the Uba1–Cdc34 complex is likely to utilize a distinct binding mode compared to previously characterized Uba1–E2 complexes[32,33] and, therefore, may provide insights into the molecular rules governing promiscuity in Uba1-E2 interactions.

After E1-E2 thioester transfer, the resulting Cdc34~Ub thioester intermediate (Cdc34~Ub, where ~ indicates a thioester) is recruited to SCF E3 ubiquitin ligases that catalyze extension of Lys48-linked polyUb chains. SCF ligases are multidomain enzymes that include Rbx1, a RING-domain protein that harbors the catalytic activity and engages E2~Ub[34–37]. E2~Ub adopts an array of conformations[38–42] and one catalytic mechanism whereby RING stabilizes E2~Ub in a closed conformation is established[43–45], though this has not been confirmed for Rbx1/Cdc34. Additionally, Cdc34 contains unique structural features that are crucial for efficient polyUb chain extension through poorly understood mechanisms, including an acidic loop insertion proximal to the active site and a long C-terminal extension. The C-terminal extension has two regions with distinct biological functions: the distal highly acidic region binds a basic canyon on SCF component Cul1 for rapid Cdc34~Ub/SCF complex formation[46,47] and the region immediately proximal to the Ubc core is implicated in self-association and non-covalent interaction with Ub thioester, Ub(t)[48–50]. Previously resolved structures of Cdc34[19,51], including one with Ub, utilized constructs that were truncated just beyond the Ubc core due to presumed disordering of the tail[49], making analysis of interactions between the tail and Ub(t) impossible. Thus, despite work spanning several decades, there are still many questions about the mechanistic roles of this C-terminal extension.

To better understand the molecular bases of Cdc34 activities, we here determine crystal structures of Cdc34 alone and in complex with Uba1, as well as a Cdc34~Ub mimetic that represents the product of E1-E2 thioester transfer. The structures illuminate conformational changes in both Cdc34 and Uba1 that are required for complex formation and reveal a unique binding mode compared to previously characterized E1-E2 complexes. Further, the Cdc34~Ub mimetic structure reveals that the proximal C-terminal extension engages Ub(t) and stabilizes Cdc34~Ub in a closed conformation. Further, these contacts are critical for Ub discharge in vitro and in cells. Collectively, our structural, biochemical, and cell-based studies contribute insights to the molecular mechanisms of Cdc34 activity in the cell.

## Results

**Overall architecture of a ScUba1-Cdc34$^{\Delta dist}$/Ub(a) complex.** A number of sequence and structural features unique to Cdc34 suggest that it engages Uba1 via a distinct binding mode during E1-E2 thioester transfer compared to other E2s. First, the physiochemical properties of amino acids in Cdc34 predicted to interact with the Uba1 UFD differ significantly compared to other E2s, including the absence of a highly conserved basic motif that has been shown to be critical for E1-E2 thioester transfer[31]. Cdc34 also harbors a short N-terminal extension proximal to the major UFD-binding region of E2s that is in position to make contacts to Uba1 that are specific to Cdc34. To reveal the roles of these unique Cdc34 features in Uba1 binding, a crystal structure of a ternary *S. cerevisiae* Uba1-Cdc34/Ub/Mg•ATP complex (Fig. 1a and Supplementary Table 1), hereafter referred to as Uba1-Cdc34, was determined using a previously described cross-linking strategy that stabilizes the complex via a disulfide bond between the Uba1 and Cdc34 catalytic cysteines[32], in vitro E1-E2 thioester transfer assays demonstrate that both CTD$^{prox}$ (residues 171–195) and CTD$^{dist}$ (residues 196–295) of the extended Cdc34 C-terminus are dispensable for thioester transfer activity (Fig. 1c and Supplementary Fig. 1) and the best diffracting crystals were obtained with a Cdc34 construct lacking CTD$^{dist}$ (Cdc34$^{\Delta dist}$). Of note, all bar graph representations of biochemical data in these studies represent quantification of product formation at the indicated timepoint, not the kinetic rate of product formation. The Uba1-Cdc34 structure was determined to 2.07 Å resolution (Supplementary Table 1) with two nearly identical copies of the complex in the crystallographic asymmetric unit.

Globally, the Uba1–Cdc34 complex exhibits a conserved architecture with the IAD and AAD engaging the Ub adenylate, Ub(a), and with Cdc34 sandwiched between the UFD and SCCH of Uba1 (Fig. 1b). Available structures of Uba1 indicate a range of conformations for the UFD that span from distal to proximal relative to the SCCH[25,32,52,53]. Previous studies suggest that the Uba1 UFD initially recruits E2 in a distal conformation and subsequently rotates into a proximal conformation to place the E2 catalytic cysteine near the Uba1~Ub thioester bond to allow transfer of Ub from Uba1 to E2. Indeed, when Cdc34$^{E1-bound}$ is docked onto the distal UFD of Uba1$^{apo}$ (PDB: 3CMM)[52], the catalytic cysteines are separated by 13 Å and would be unable to participate in thioester transfer (Fig. 1d, *left*). Comparison of Uba1$^{Cdc34-bound}$ to Uba1$^{apo}$ reveals a 20° rotation of the UFD proximally (Fig. 1d, *middle*) to bring the catalytic cysteines of Uba1 and Cdc34 together for Ub thioester transfer (Fig. 1d, *right*).

**Molecular recognition of Cdc34 by Uba1.** Analysis of the Uba1-Cdc34 structure reveals three networks of contacts at distinct

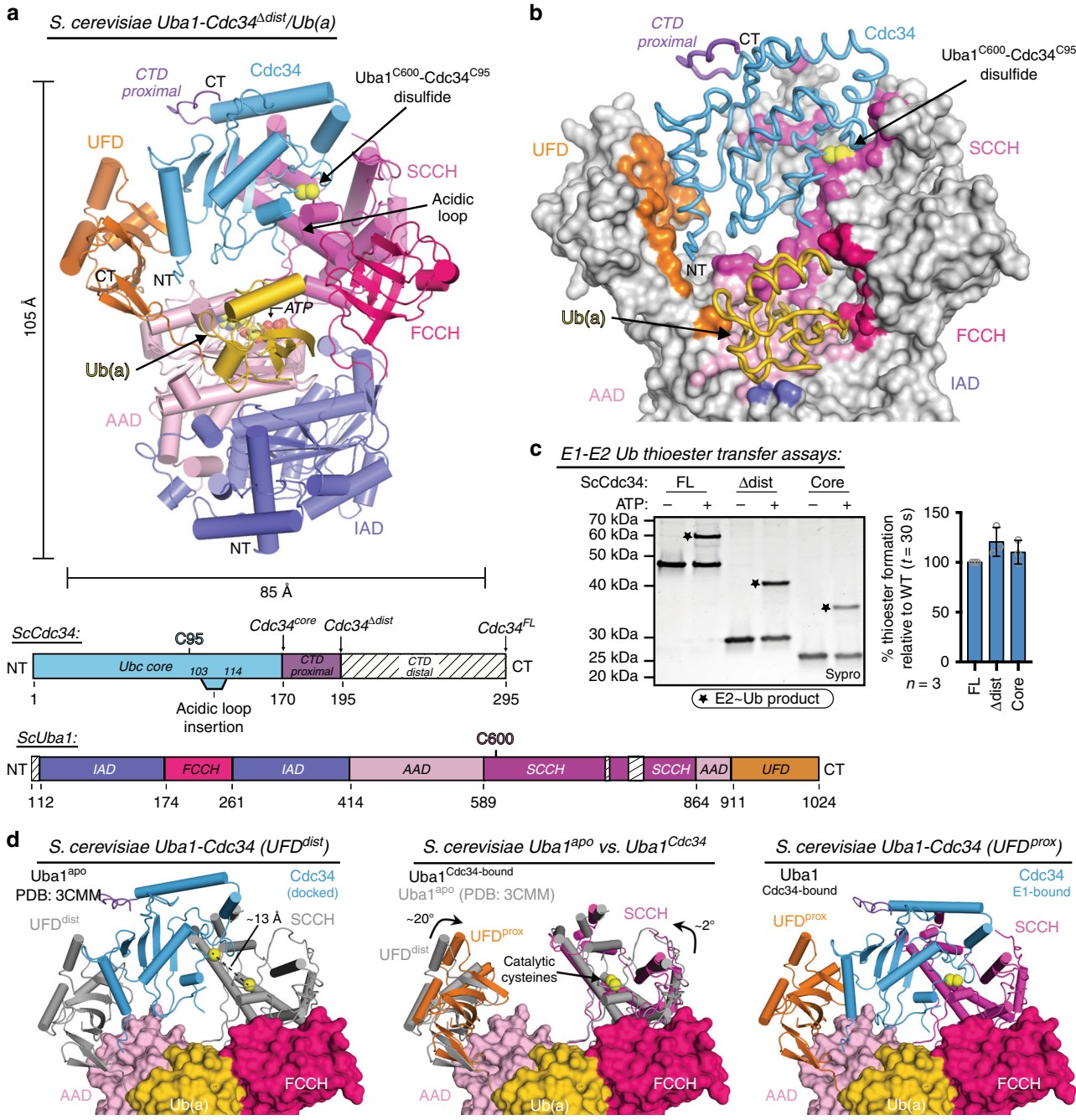

**Fig. 1** Overall architecture of a ScUba1-Cdc34$^{\Delta dist}$/Ub(a) complex. **a** *Top*, overall structure of *S. cerevisiae* Uba1 in complex with truncated *S. cerevisiae* Cdc34, Cdc34$^{\Delta dist}$ (blue), and ubiquitin adenylate, Ub(a) (gold), with Uba1 domains colored and labeled. Catalytic cysteines are indicated by yellow spheres. *Bottom*, schematic representations of the ScUba1 and ScCdc34 constructs with domains colored as above. **b** Uba1-Cdc34 structure with Uba1 surface representation in gray with only contacting residues colored. Cdc34$^{\Delta dist}$ and Ub(a) are represented as worms. **c** E1-E2 thioester transfer assays for Cdc34 variants with boundaries as indicated in schematic, and with normalized quantification of E2 thioester formed represented as mean ± SD with three independent replicates shown as gray circles, *right*. Stars indicate the E2~Ub thioester product. Source data are provided as a Source Data file. **d** *Left*, Cdc34$^{E1-bound}$ modeled onto Uba1$^{apo}$ UFD (gray) in open conformation (PDB:3CMM); distance between catalytic cysteines is indicated by a dashed line. *middle*, ScUba1$^{Cdc34-bound}$/Ub(a) as in **a** with ScUba1$^{apo}$/Ub(a) superimposed in gray. Arrows indicate rotation of the UFD and SCCH when bound to Cdc34. *Right*, ScUba1-Cdc34$^{\Delta dist}$/Ub(a) complex structure colored as in **a**

interfaces: Cdc34/UFD, Cdc34/SCCH, and a tripartite interaction between Cdc34, Ub(a), and the Uba1 crossover loop that connects the AAD to the SCCH (Fig. 2a). One of the most prominent features of the UFD is a conserved acidic patch that includes Glu1004, Asp1008, Asp1014, and Glu1016. At the heart of the Cdc34/UFD interface lies a network of salt bridges between the UFD acidic patch and basic residues of Cdc34 (Fig. 2d).

Specifically, Arg14 and Arg17 of Cdc34 helix A (hA) form salt bridges with Asp1014 and Glu1016 on the UFD, as well as a water-mediated hydrogen bond with Asp1008 (Supplementary Fig. 2). Introduction of charge reversal mutations at any of these positions on Uba1 or Cdc34 results in a significant loss of thioester transfer activity (Fig. 2e). Moreover, combining Cdc34 (R14E/R17E) and Uba1 (D1014R/E1016R) charge reversal

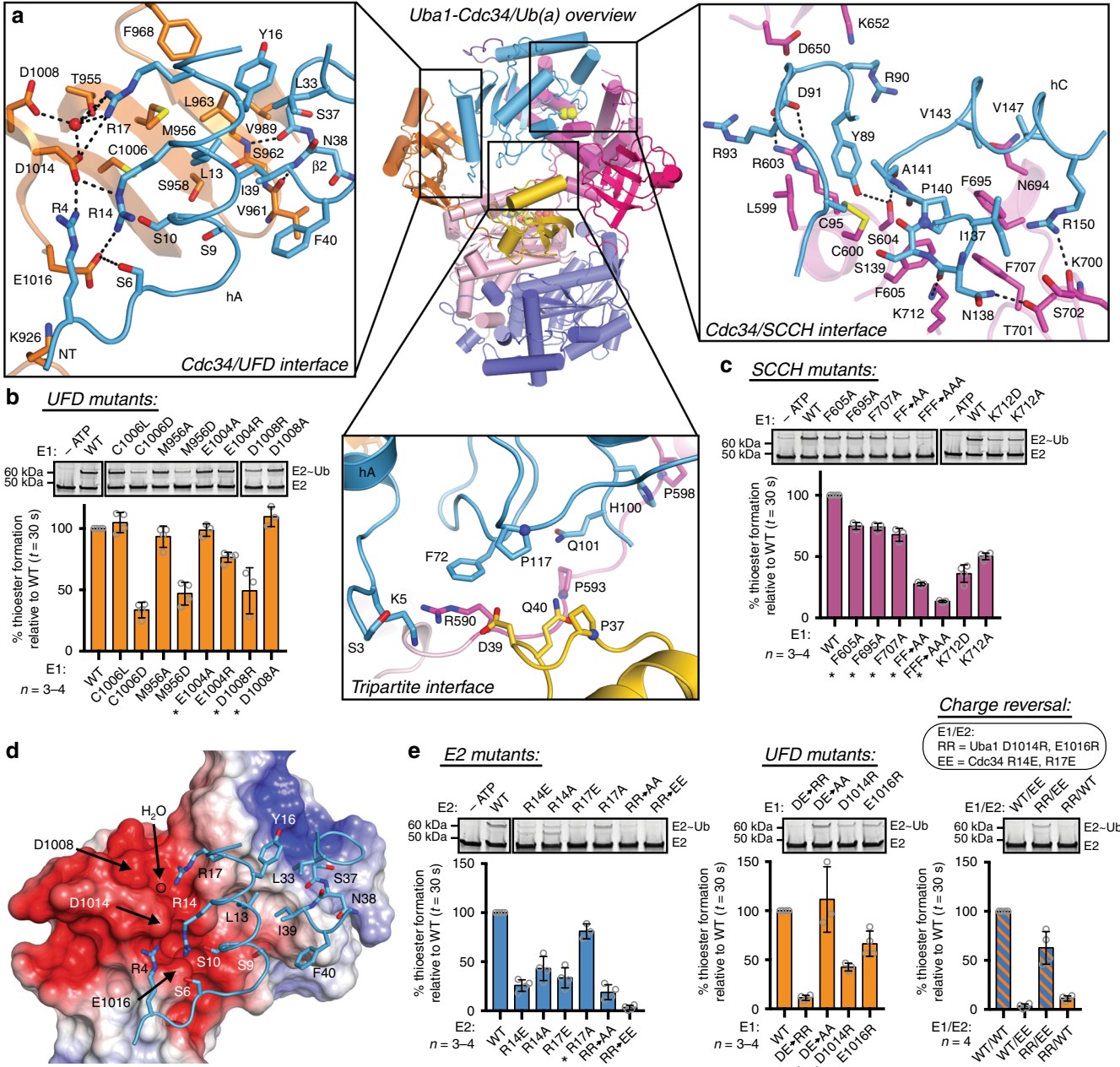

**Fig. 2** Combinatorial recognition of Cdc34$^{\Delta dist}$ by ScUba1. **a** *Top middle*, overall structure of Uba1-Cdc34$^{\Delta dist}$/Ub as in Fig. 1a. *Top left*, network of contacts between Uba1 UFD and Cdc34 with involved residues shown as sticks with red oxygen atoms, blue nitrogen atoms, and yellow sulfur atoms. Hydrogen bonds are indicated by dashed lines. *Top right*, interaction network between Cdc34 and Uba1 SCCH. *Bottom middle*, tripartite network between Cdc34, Ub (a), and Uba1 crossover loop. **b**, **c** E1-E2 thioester transfer assays of the indicated mutants for Uba1 UFD (**b**) and SCCH (**c**). Data are represented by mean ± SD with representative replicates labeled above and individual replicates shown as gray circles. Biochemical assays have four independent replicates, unless indicated by an asterisk when there are three replicates. **d** Electrostatic surface of Uba1 UFD with Cdc34 contacting residues overlaid as in **a**. **e** E1-E2 thioester transfer assays of the indicated mutants for Cdc34 (E2), Uba1 UFD, and a charge reversal rescue assay with mutant Uba1 and Cdc34, represented as in **b**, **c**. Source data are provided as a Source Data file

mutants rescues a near complete loss of activity when either mutant was tested with a WT partner (Fig. 2e). This confirms the critical importance of these salt bridge interactions in the proper positioning of Cdc34 for E1-E2 thioester transfer. Aligning with these findings, previous studies have shown a reduction in *S. cerevisiae* viability when Arg14, Arg17, and Glu18 are all mutated to Ala[54].

At the periphery of the interactions described above is a supporting network of contacts between the UFD and Cdc34 hA that includes UFD residues Lys926, Thr955, Met956, Leu963,

Phe968, Cys1006, and Val1015; and hA residues Ser6, Ser10, Leu13, and Tyr16. Ser3 and Arg4 from the Cdc34 N-terminal extension also reach up to contact Asp1014, Val1015, and Glu1016 (Fig. 2a). Additionally, the Uba1 β29-strand residues Val961 and Leu963 form backbone hydrogen bonds with β1-β2 loop and β2-strand residues Ser37 and Ile39. Charge mutation of the supporting Uba1 residues Met956, Glu1004, Cys1006, and Asp1008 reduces efficiency of thioester transfer while more conservative Ala substitutions exhibit wild-type activity (Fig. 2b). This pattern suggests that, while these residues are buried at the

interface of hA and the UFD, they are not individually critical for optimal positioning of Cdc34 for E1-E2 thioester transfer.

On the other side of Cdc34, the helix C/helix D (hC/hD) surface forms an extensive network of contacts with the SCCH around the catalytic cysteines (Fig. 2a, right). Proximal to the active site cysteines, Cdc34 Tyr89 contacts Uba1 Leu599, Cys600, Arg603, and Ser604, including a hydrogen bond to Ser604 (Fig. 2a, right). Additionally, Cdc34 Ser139 and Ala141 contact Cys600 and Ser604. To the left of the catalytic cysteines, Cdc34 Asp91 forms a salt bridge with Arg603 of Uba1. To the right of the catalytic cysteines, Cdc34 Ile137 contacts Uba1 Ser702, and Asn138 creates a network of contacts with Thr701 and Lys712, including two hydrogen bonds. Highlighting its importance at this interface, disruption of the Lys712 hydrogen bond by mutation to Asp or Ala results in a moderate loss of thioester transfer activity (Fig. 2c). Cdc34 Pro140 inserts into a hydrophobic pocket on Uba1 formed by three phenylalanine residues 605, 695, and 707, mutation of which results in a loss of E1-E2 thioester transfer activity (Fig. 2c). Finally, a tripartite network of contacts between Cdc34, Ub(a), and the Uba1 crossover loop is observed wherein Cdc34 Phe72 contacts Uba1 Arg590; Ub Gln40 contacts Uba1 Pro593; and Ub Asp39 contacts Uba1 Arg590 and Cdc34 Phe72 (Fig. 2a, bottom middle). Thus, Cdc34 and Uba1 engage in extensive contacts at three distinct interfaces for productive complex formation and transthiolation.

**A flexible Uba1 architecture accommodates E2-binding modes.** Comparison of the Uba1–Cdc34 complex to previously determined Uba1-Ubc4 and Uba1-Ubc15 structures[32,33] reveals three distinct Uba1-binding modes. Superimposition of the UFDs of these structures unmasks significant differences in how hA of each E2 engages the UFD (Fig. 3a, middle). Specifically, hA of Cdc34 rotates 7° and translates 0.9 Å toward the bottom of the UFD with respect to Ubc4. With respect to Ubc15, hA of Cdc34 rotates 13° and translates 1 Å toward the top of the UFD (Fig. 3a, left; 3b). These seemingly subtle differences at the hA/UFD interface result in significantly different positioning of the active sites and SCCH-binding surfaces of the E2s relative to the UFD (Fig. 3a, middle). Highlighting this, the catalytic cysteine of Cdc34 has translated 3.7 Å compared to Ubc4 and 6.9 Å compared to Ubc15 (Fig. 3a, right). In contrast, superimposition by the SCCH reveals that the E2/SCCH interface is better conserved than the UFD/E2 interface (Fig. 3c). The overall orientation of the E2s relative to the SCCH are similar and the catalytic cysteines remain in proximity via a common network of contacts (Supplementary Fig. 3).

A variable E2/UFD interface that results in positioning of E2 catalytic cysteines up to 7 Å apart from each other (Fig. 3a, b) together with a conserved E2/SCCH interface (Fig. 3c) raises the question of how Uba1 and E2 active sites can come together as required for E1-E2 thioester transfer. Superimposition by the Uba1 adenylation domains, which are the rigid body of the enzyme[52], reveals that the architecture of Uba1 in each of the structures is distinct, with the UFD and SCCH exhibiting relative differences in rotation of 5–9° and 4–6°, respectively (Fig. 3d). Altogether, these distinct domain rotations are the mechanism by which E1 and E2 active sites are brought into proximity during E1-E2 thioester transfer despite differences in E2/UFD-binding mode.

**Structural plasticity at the E2/UFD interface.** Comparison of the Uba1-Cdc34, Uba1-Ubc4, and Uba1-Ubc15 E2/UFD interfaces reveals distinct networks of contacts (Fig. 4a, top) that reflect the markedly different angles at which E2 hAs engage the UFD (Fig. 3a). One clear difference at this interface is the network of

interactions mediated by the Uba1 UFD acidic patch. Previous modeling and biochemical experiments led to the prediction that the acidic patch interacts with a highly conserved three residue basic motif on E2 hA that corresponds to ⁴KRINR⁸ in Ubc4[31,52]. Interestingly, although Cdc34 contains only the last basic residue in the motif, Arg14, Cdc34 interacts with a larger surface of the acidic patch than both Ubc4 and Ubc15 (Fig. 4a, middle). This is due to an additional basic residue proximal to Arg14, Arg17, which, facilitated by a distinct hA orientation, leads to extensive contacts between Arg14/Arg17 and the top of the UFD acidic patch that are critical for proper Uba1-Cdc34 thioester transfer in vitro (Fig. 2d, e) and Cdc34 activity in cells[54]. In contrast, Ubc4 lacks a basic residue equivalent to Arg17 and the orientation of hA is such that the top of the UFD acidic patch makes no contacts to Ubc4 (ref. [32]) (Fig. 4a, top). Ubc15 also contains only the last basic residue of the basic motif, Lys12, and Lys15 that corresponds to Cdc34 Arg17. Contrary to the short-range salt bridge interactions between the acidic patch and well-ordered Cdc34 Arg14/Arg17, Ubc15 Lys12, and Lys15 are poorly ordered and participate in longer-range electrostatic interactions with the acidic patch (Fig. 4a, top). This is due to both the shorter lysine side chains and the presence of an acidic residue, Glu7, that necessitates a downward shift of Ubc15 hA to avoid electrostatic repulsion between Glu7 and the UFD acidic patch, positioning Lys12 and Lys15 out-of-range for potential salt bridge interactions (Fig. 4a, top).

To further probe the plasticity of the UFD/E2 interface, residues in Ubc4 and Ubc15 were mutagenized to more closely resemble Cdc34 (Fig. 4b and Supplementary Fig. 4). In Ubc4, mutating Lys4 to Cdc34 Ser10 resulted in a 2-fold loss of activity, while mutation of Ala11 to Cdc34 Arg17 had no significant effect on thioester transfer activity (Fig. 4b and Supplementary Fig. 4). Interestingly, the double mutation (KA→SR) rescued K4S activity loss, suggesting that Ubc4 may be able to interact with the acidic patch similarly to Cdc34. In Ubc15 a K12A mutant exhibits a 5-fold loss of activity whereas K15A has a mild increase in activity (Fig. 4b and Supplementary Fig. 4). Surprisingly, a K12A/K15A double mutation results in a near complete loss of Ubc15 activity indicating that, while it is otherwise dispensable, Lys15 becomes necessary in the absence of Lys12. This suggests that the UFD is able to compensate for loss of important E2 contacts by reorganizing to form favorable contacts with other residues in proximity, further supporting structural plasticity at the UFD/E2 interface as the mechanism underlying promiscuity in E1-E2 interactions.

Outside of the UFD acidic patch, a conserved set of UFD residues including Met956 and Cys1006 engage different sets of E2 hA residues due to the different orientations of E2 hA and sequence diversity among E2s at this region (Figs. 3a and 4). Another major region of variability at the E2/UFD interface involves the variable N-terminal extensions of E2s (Fig. 4a). While Cdc34's extension projects outward between the UFD and AAD making additional contacts to the UFD acidic patch and AAD, the Ubc15 N-terminal extension takes a more downward path into the AAD, contacting a region of the UFD below the acidic patch and making far more contacts to the AAD[33] (Fig. 4a). Conversely, Ubc4 does not possess an N-terminal extension and this is highlighted by fewer contacts to the UFD and no contacts to the AAD (Fig. 4a). More than 20 active human E2s contain variable N-terminal extensions, and this analysis suggests they may engage the E1 in distinct ways.

In summary, these analyses reveal that the Uba1 UFD utilizes a conserved core of residues to build distinct networks of contacts with divergent E2 hA sequences. Additionally, variable regions outside of this conserved UFD core, including the AAD, are engaged to accommodate unique N-terminal extensions.

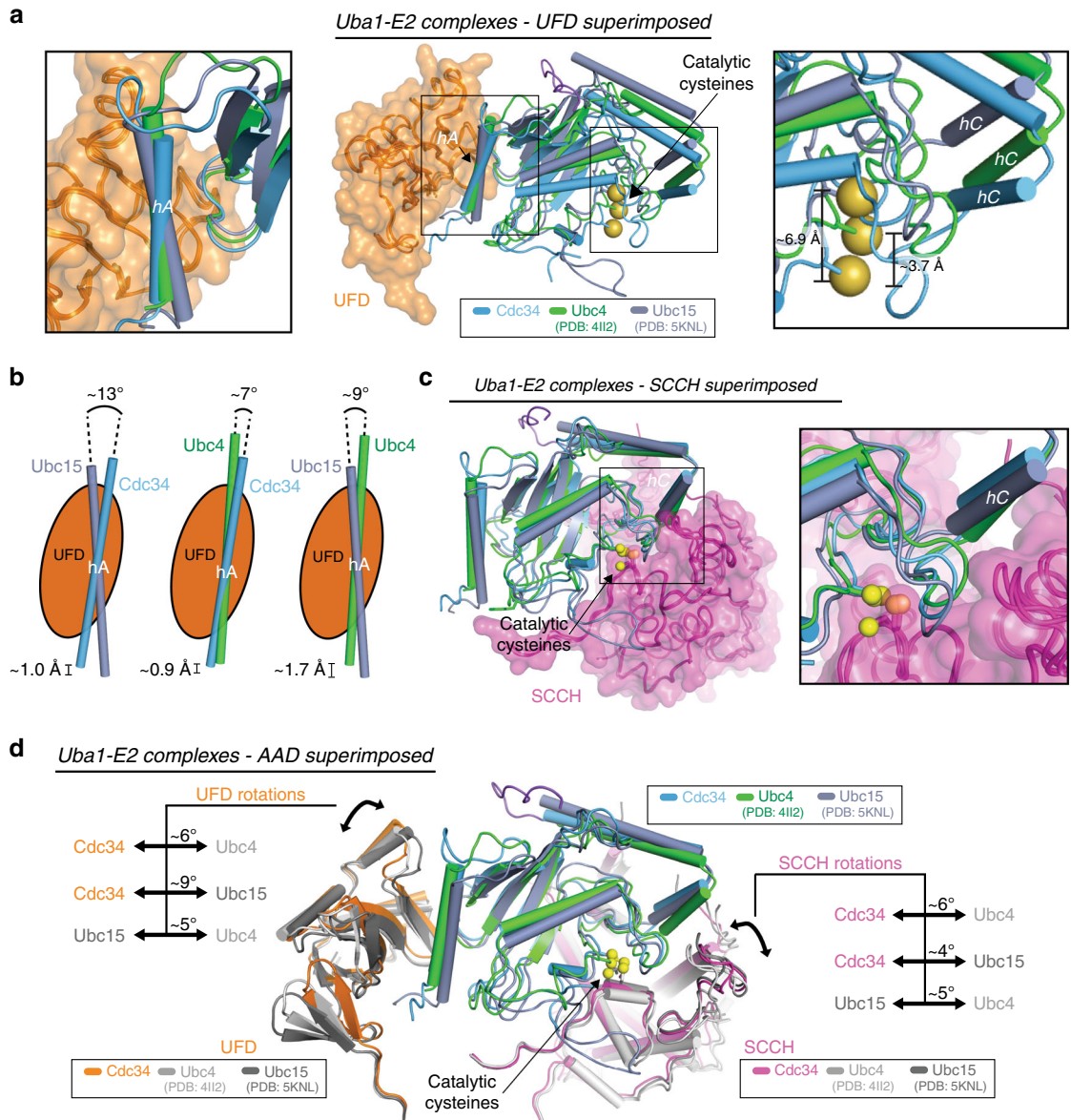

**Fig. 3** Distinct Uba1 architectures accommodate different E2-binding modes. **a** *Middle*, superimposition of Uba1-Cdc34, Uba1-Ubc4 (PDB:4II2), and Uba1-Ubc15 (PDB:5KNL) complexes by the UFD. Cdc34 is colored blue, Ubc4 is green, Ubc15 is gray. hA/UFD interface and catalytic cysteine positioning are highlighted by black boxes. *Left*, zoomed in, head-on view of E2 hA binding to Uba1 UFD. *Right*, zoomed in view of catalytic cysteine positions. **b** Cartoon schematic of difference in angle of hA bound to Uba1 for each E2 pair, as in **a**. **c** *Left*, superimposition of Uba1–E2 complexes by the SCCH with the E2/SCCH interface highlighted by a black box. *Right*, zoomed in view of the E2/SCCH interface. **d** Superimposition of Uba1–E2 complexes by the AAD with the E2, UFD, and SCCH shown. Relative differences in rotation for the UFD (*left*) and SCCH (*right*) are indicated by arrows

Altogether, this results in an altered orientation of each E2 hA relative to the UFD and necessitates distinct rotations of both the UFD and SCCH in order to bring E1 and E2 catalytic cysteines in proximity for productive thioester transfer in the context of different E1-E2-binding modes.

**Cdc34 conformational changes accompany Uba1 binding**. To investigate potential Cdc34 conformational changes that accompany Uba1 binding, a 1.7 Å structure of ScCdc34$^{\Delta dist}$ alone, Cdc34$^{apo}$, was determined (Fig. 5a and Supplementary Table 1). Comparison of Cdc34$^{apo}$ and Cdc34$^{E1-bound}$ reveals two conformational changes that accompany Uba1 binding, both within E1-interacting regions. First, compared to Cdc34$^{E1-bound}$, hA of Cdc34$^{apo}$ (hA$^{apo}$) is extended N-terminally by 5 residues, beginning at Ser3 instead of Ala8. When bound to Uba1, this

extended region of hA melts but remains ordered and projects away from the helix (Figs. 2a, *left*, and 5a; and Supplementary Fig. 5). The second conformational change occurs on the opposite side of the E2, where hC of Cdc34$^{apo}$ (hC$^{apo}$) is extended N-terminally by one residue and sits at a different angle compared to hC$^{E1-bound}$ (Fig. 5a and Supplementary Fig. 5). Modeling Cdc34$^{apo}$ onto the Uba1–Cdc34 complex reveals a mechanistic role for Cdc34 conformational changes that accompany Uba1 binding (Fig. 5b, *middle*). First, the extended hA$^{apo}$ clashes with Uba1 AAD, including backbone and sidechain atoms of Ser3, Arg4, Lys5, and Ser6, whereas the melted region of hA$^{E1-bound}$ packs into a small pocket between the UFD and AAD and makes additional contacts to Uba1 (Fig. 5b, *left*). Second, the extension and altered angle of hC$^{apo}$ positions Val143 such that it clashes with the SCCH, whereas Cdc34$^{E1-bound}$ Val143 is shifted away from the SCCH and makes productive contacts with Uba1

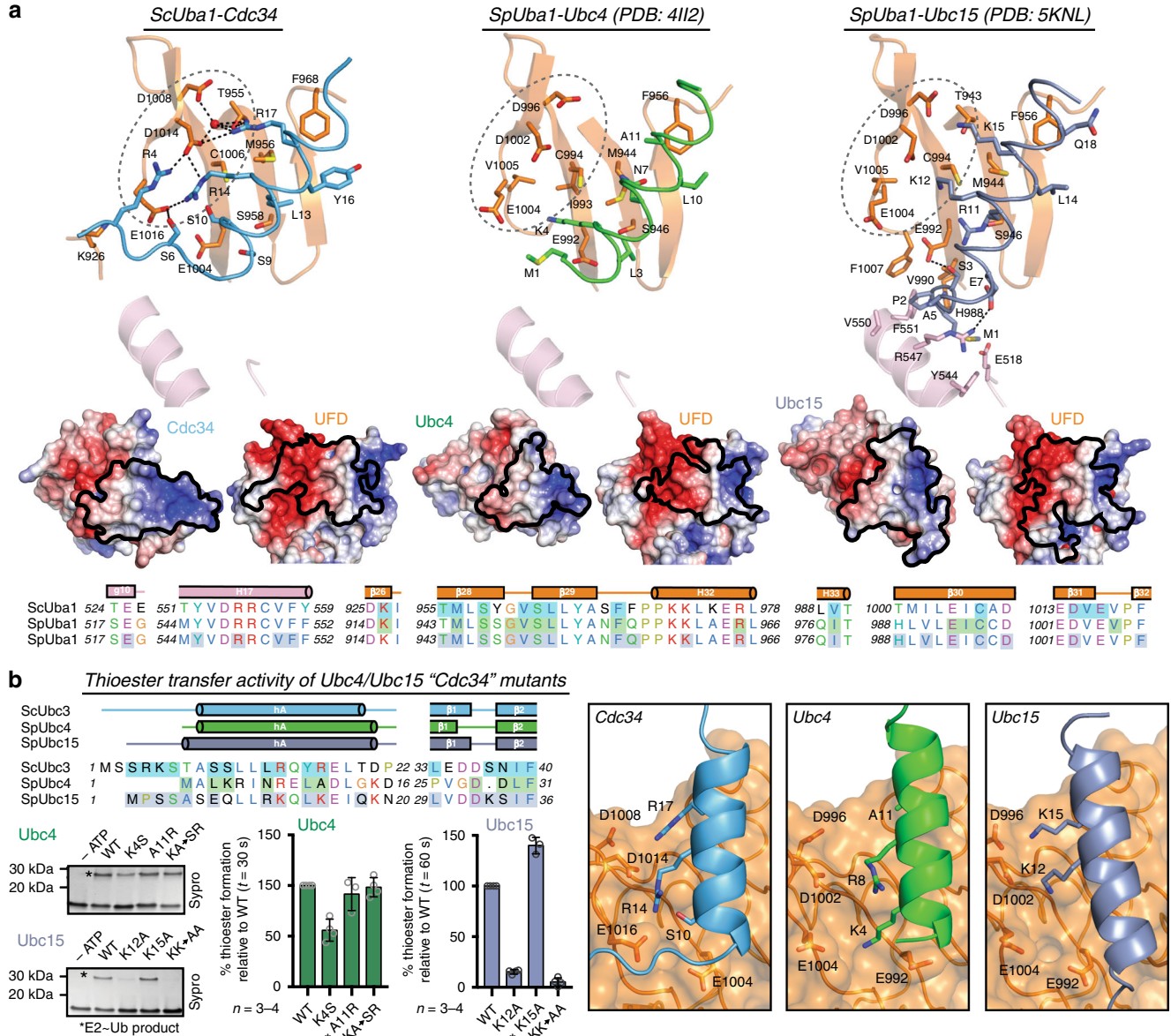

**Fig. 4** The E1/Cdc34-binding mode is achieved via a unique network of contacts. **a** *Top*, interface between E2 and Uba1 UFD with involved residues shown as sticks for Cdc34 (*left*), Ubc4 (*middle*), and Ubc15 (*right*); colored as in Fig. 3. Dashed circles indicate UFD acidic patch binding region. *Middle*, open-book electrostatic surfaces of the E2/UFD interface with UFD footprint mapped onto E2 (*left*) and E2 footprint mapped onto UFD (*right*) for each E2 above. *Bottom*, sequence alignment of ScUba1 and SpUba1 AAD and UFD with corresponding secondary structure cartoon shown above sequence. Shaded boxes indicate Uba1 residues that interact with the corresponding E2: Cdc34 (blue), Ubc4 (green), and Ubc15 (gray). **b** *Top left*, sequence alignment of E2 UFD/AAD-interacting region with secondary structure cartoon and interacting residues shaded as in **a**. *Bottom left*, E1-E2 thioester transfer assays of the indicated mutants for Ubc4 and Ubc15. Data are represented by mean ± SD with representative replicates labeled to the left and individual replicates shown as gray circles. Biochemical assays have four independent replicates, unless indicated by an asterisk when there are three replicates. Source data are provided as a Source Data file. *Right*, interface between UFD acidic patch and E2 hA residues corresponding to Cdc34 Arg14/Arg17

Asn694 and Phe695 (Figs. 2a, *right*; and 5b, *right*). Thus, these findings suggest that the observed Cdc34 conformational changes that accompany Uba1 binding are important for productive Uba1–Cdc34 complex formation.

**The Cdc34 CTDprox extension is involved in Ub discharge**. The Cdc34 C-terminal extension is required for efficient Cdc34 function with CTDprox and CTDdist playing distinct roles[46–50,55]. While a role for CTDdist in facilitating rapid association of the Cdc34~Ub/SCF complex by binding to a basic canyon on Cul1 has been demonstrated[46,47,56], the molecular basis by which

CTDprox promotes Cdc34 activity is less clear. Although nuclear magnetic resonance studies suggest interactions between CTDprox and Ub(t)[49,50], the structural basis for the interaction and the mechanism for increased Cdc34 activity are unknown. Analysis of our Cdc34apo and Cdc34E1-bound structures reveals additional ordering of CTDprox beyond what has been observed in previously determined Cdc34 structures[19,51] (Fig. 5a, c). This additional ordered CTDprox region engages in extensive contacts with the backside of Cdc34 and projects toward the anticipated location of Ub(t) in the closed conformation (Fig. 5c). These findings along with evidence for CTDprox participation in non-covalent contacts to Ub(t)[49,50] led us to hypothesize that CTDprox extends

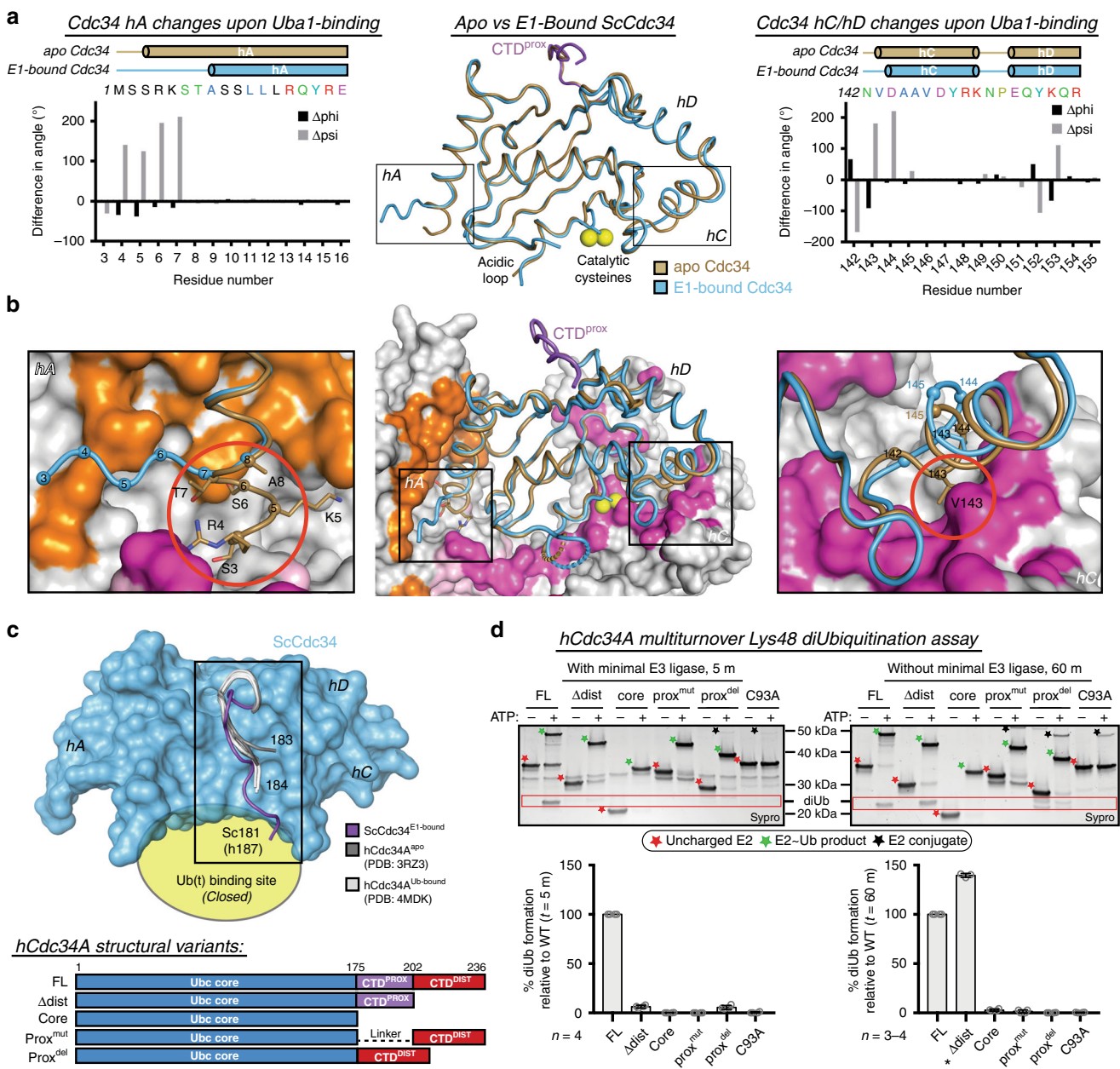

**Fig. 5** Structural changes in key Cdc34 elements accompany E1 binding. **a** *Middle*, *apo* (tan) and E1-bound (blue) Cdc34 superimposed using the conserved core of the protein. hA and hC conformational changes are highlighted by black boxes. Differences in phi and psi angles between Cdc34[E1-bound] and Cdc34[apo] are shown as bar graphs for hA (*left*) and hC/hD (*right*). Cartoons above the graphs indicate secondary structure boundaries for Cdc34[apo] (tan) and Cdc34[E1-bound] (blue). **b** *Middle*, Cdc34[apo] superimposed as in **a** onto Uba1–Cdc34 complex. Uba1 is colored as in Fig. 1 and Cdc34 conformational changes are highlighted as in **a**. Zoomed in view of Cdc34 hA (*left*) and hC (*right*) with clashes between Uba1 and Cdc34[apo] indicated by red circles. **c** *Top*, top-down view of CTD[prox] of ScCdc34[E1-bound] (purple), hCdc34A[apo] (PDB:3RZ3, dark gray), and hCdc34A[Ub-bound] (PDB:4MDK, light gray) superimposed as worms with ScCd34 core shown as blue surface, and closed Ub(t)-binding site indicated by a yellow oval. *Bottom*, cartoon schematic of hCdc34A global mutants used for K48-diUb assays. **d** K48-diUb assays for the indicated mutants with (*left*) and without (*right*) minimal E3 ligase. Data are represented by mean ± SD with individual replicates shown as gray circles. Biochemical assays have four independent replicates, unless indicated by an asterisk when there are three replicates. Representative replicates are labeled above and annotated with stars for clarity. diUb product is indicated by a label and red box. Source data are provided as a Source Data file

beyond the Cdc34 backside to engage Ub(t) in a Cdc34~Ub complex. Although a Cdc34/Ub non-covalent complex structure exists[51], the Cdc34 construct lacked the full CTD[prox] and, thus, a Cdc34~Ub structure that includes the CTD[prox] might capture this putative CTD[prox]/Ub(t) interaction.

To first test the roles of the CTD[prox] and CTD[dist] in Ub(t) discharge, several human Cdc34A mutant constructs were cloned

and subjected to a modified multiple-turnover Lys48-linked diubiquitination (K48-diUb) assay[57] (Fig. 5d and Supplementary Fig. 5). Briefly, Cdc34 is continuously loaded with Ub by Uba1, with or without minimal SCF E3, in the presence of excess ubiquitin to allow formation of Lys48-linked diUb by Cdc34. The specificity of this assay for Lys48-linked diUb is demonstrated by utilizing K48R Ub in place of WT Ub (Supplementary Fig. 6). The tested variants

include: Cdc34$^{\Delta dist}$ containing full CTD$^{prox}$, Cdc34$^{core}$ containing only the Ubc core domain of the protein, and prox$^{mut}$ and prox$^{del}$ where the CTD$^{prox}$ region is either mutated to a flexible linker or deleted, respectively (Fig. 5c, *bottom*). Interestingly, in the absence of E3, only WT and Cdc34$^{\Delta dist}$ were able to produce K48-diUb and produced similar amounts (Fig. 5d, *right*). In the presence of E3, only WT Cdc34 was capable of producing K48-diUb under the tested assay conditions, suggesting that the CTD$^{prox}$ and CTD$^{dist}$ regions are both critical for productive E3 interaction and catalysis of Cdc34 Ub(t) discharge (Fig. 5d, *left*). At a longer timepoint with E3, Cdc34$^{\Delta dist}$ appears to behave as though there is no E3 in the reaction, suggesting that the deficiency observed at 5 m is due only to the lack of E3 catalysis (Supplementary Fig. 5). Under these assay conditions, E2~Ub formation is not affected by these mutations, indicating that a defect in E1-E2 thioester transfer does not contribute to downstream effects on K48-diUb formation (Fig. 5d and Supplementary Fig. 5). Thus, CTD$^{prox}$ is necessary and sufficient for the inherent ability of Cdc34 to discharge Ub while both the CTD$^{prox}$ and CTD$^{dist}$ are necessary for productive E3 catalysis. Comparatively, equivalent mutants of yeast Cdc34 display a similar phenotype in the presence of E3 catalysis (Supplementary Fig. 5). Additionally, under E3-free conditions the yeast Cdc34 mutants also exhibit a reduction in activity, though it is less severe (Supplementary Fig. 5). Altogether, these observations support the hypothesis that the CTD$^{prox}$ contacts Ub(t)[49] and is consistent with previous modeling studies that suggested the acidic CTD$^{dist}$ binds a basic canyon on Cul1[46].

**Cdc34 CTD$^{prox}$ locks Ub(t) in the closed conformation**. To elucidate the mechanism by which CTD$^{prox}$ enhances Cdc34 activity, a crystal structure of a Cdc34~Ub(t) complex that includes a complete CTD$^{prox}$ is needed. To overcome the inherent instability of the Cdc34~Ub thioester bond, a human Cdc34B$^{\Delta dist}$~Ub mimetic (Cdc34-Ub, where - indicates an isopeptide linkage) was produced wherein a stable isopeptide bond between Cdc34 active site Cys→Lys mutant and the Ub(t) C-terminal glycine replaces the labile thioester bond, using a modification of a previously described protocol[45,58]. A 1.8 Å structure of Cdc34-Ub was determined in the presence of Cdc34 inhibitor, CC0651, which previous studies show stabilizes the interaction between Ub(t) and the closed binding surface on Cdc34 (ref. [51]) (Fig. 6a, b and Supplementary Table 1). Analysis of the overall Cdc34-Ub structure shows that, as anticipated, the complex adopts the closed conformation with Ub(t) engaging the crossover helix (hB) of Cdc34 (Fig. 6a, b). The Cdc34 acidic loop insertion is ordered in this structure, but aside from contacts to Arg74, it does not extensively interact with Ub(t) and is likely ordered due to crystal contacts. Additionally, the C-terminus of Ub is completely ordered through the isopeptide bond with Cdc34 Lys93 (Supplementary Fig. 6). Importantly, comparison of the Cdc34-Ub structure to the previously determined non-covalent hCdc34A/Ub structure (PDB: 4MDK) reveals that eight additional residues of CTD$^{prox}$ become ordered and engage in a network of contacts with Ub(t) that fortify the Cdc34/Ub closed interface (Fig. 6a, *right*; 6b, *right*). Previous studies indicate that the closed conformation of E2~Ub is the active form of the complex and one key role of canonical RING E3 is to lock E2~Ub in the closed conformation that is primed for Ub discharge[43]. Thus, analysis of our Cdc34-Ub complex suggests that the structural basis by which CTD$^{prox}$ enhances Cdc34 activity is stabilization of the Cdc34~Ub closed conformation.

Detailed analysis of the Cdc34-Ub structure reveals that CTD$^{prox}$ engages in an extensive network of intramolecular interactions with the backside of Cdc34 that is known to bind non-covalent Ub to allosterically enhance activity in a subset of

E2s[59] (Fig. 6b and Supplementary Fig. 6). The network of contacts between CTD$^{prox}$ and the Cdc34 core involves Val181, Val183, Pro184, and Thr185 of CTD$^{prox}$ and Phe46, Pro49, Glu54, Gly55, Tyr57, and Ala176 of the core. Val181 contacts both Phe46 and Tyr57 of the core and forms a backbone hydrogen bond to Ala176. Val183 contacts Phe46, Gly55, and Tyr57, while Pro184 primarily contacts Phe46 (Fig. 6b, *right*). Finally, Thr185 contacts Pro49 and Glu54 of the core. Single mutations of any of the central residues at this interface: CTD$^{prox}$ Val181, Val183, and Pro140, as well as core Phe46 and Tyr57, results in near total loss of K48-diUb activity in the presence of E3 and severe reduction of activity in the absence of E3 (Fig. 6c, *top* and *middle*). This highlights the importance of this intramolecular network of interactions for efficient Ub discharge by Cdc34, with or without E3 catalysis.

The intramolecular interactions between Cdc34 CTD$^{prox}$ and the backside of the Cdc34 core guide the CTD$^{prox}$ toward the closed Ub(t) where intermolecular interactions occur. These interactions involve a short α-helix (hE) in the additionally ordered region of CTD$^{prox}$. Tyr190 of Cdc34 CTD$^{prox}$ inserts into a composite pocket formed by residues from both Ub(t) and the Cdc34 core. Specifically, Tyr190 engages in interactions to Ub(t) Ala46 and Gly47 and Cdc34 core residues Glu26, Gly27, Phe46, and Pro49. Additionally, CTD$^{prox}$ Cys191 contacts Ub(t) Ala46, while CTD$^{prox}$ Leu187 contacts Ub(t) Phe45 and Ala46 and participates in longer-range contacts to Ub(t) Tyr59 and Asn60 (Fig. 6b, *right*). In K48-diUb assays, mutation of Tyr190 to either Asp or Ala completely abolishes K48-diUb formation in the presence of E3 and nearly abolishes activity in the absence of E3 (Fig. 6c, *top* and *middle*). Further, mutation of Cdc34 Leu187 and Cys190 also significantly impairs K48-diUb activity in the presence and absence of E3 (Fig. 6c, *top* and *middle*). These data suggest a key mechanistic role for Cdc34 CTD$^{prox}$ in stabilizing the Cdc34~Ub closed conformation that is primed for Ub(t) discharge.

Previous studies of Cdc34 and other E2s indicate that there are two surfaces on Cdc34 that play an important role in polyUb chain formation: one surface that interacts with Ub(t) (the donor Ub, Ub$^D$) to prime the active site for catalysis and a second that is involved in proper positioning of the acceptor Ub (Ub$^A$) lysine for attack of the Cdc34~Ub$^D$ thioester bond[60-64]. Thus, to probe the importance of interactions between Cdc34 CTD$^{prox}$ and Ub(t) from the Ub(t) side of the complex, an adapted version of a previously described single turnover diUb formation assay[62] was utilized. Unlike the multiturnover K48-diUb assay, the single turnover K48-diUb assay can distinguish between Ub residue roles as a donor versus an acceptor. Additionally, as for the multiple-turnover assay, the single turnover assay specifically produces Lys48-linked diUb (Supplementary Fig. 6). Consistent with the Cdc34-Ub structure, F45D, A46D, and G47D Ub mutants each exhibit a significant decrease in K48-diUb formation when used as Ub$^D$ under single turnover assay conditions due to disruption of interactions with CTD$^{prox}$ (Fig. 6c, *bottom*). As might be expected based on the proximity to Lys48, F45D, A46D, and G47D mutants also exhibit a significant loss of K48-diUb activity when used as Ub$^A$, likely due to disruption of contacts required for proper positioning of Ub$^A$ Lys48 during attack of the Cdc34~Ub$^D$ thioester bond (Fig. 6c, *bottom*). Altogether, these data further validate the importance of the identified interactions between the Ub(t) and Cdc34 CTD$^{prox}$ in fortifying the Cdc34~Ub closed conformation (Fig. 6b) and confirm the importance of this loop in positioning Ub$^A$ Lys48 for nucleophilic attack during K48-polyUb formation.

**CTD$^{prox}$/Ub contacts are important for Cdc34 action in cells**. We next examined the functional importance of Cdc34 CTD$^{prox}$ interactions with Cdc34 core and Ub(t) in mammalian cells;

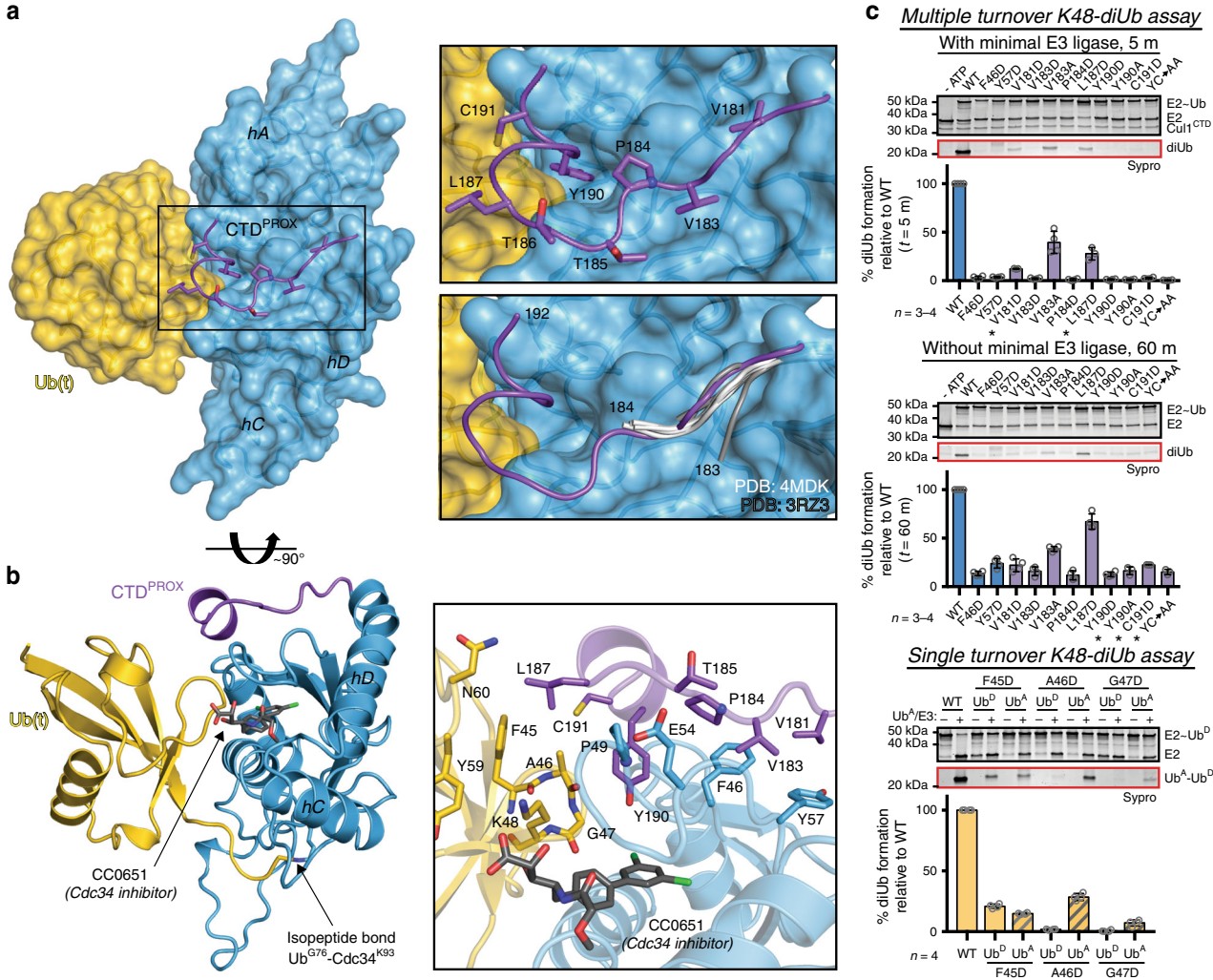

**Fig. 6** An ordered Cdc34 CTDprox extension is involved in Ub discharge. **a** *Left*, crystal structure of hCdc34BΔdist conjugated to Ub (gold) at active site Cys to Lys mutation in presence of CC0651 inhibitor, shown as in Fig. 5c but rotated 90°. *Top right*, zoomed in view of Cdc34 CTDprox trajectory over the Ubc core with contacting residues shown as sticks. *Bottom right*, zoomed in view of CTDprox as in Fig. 5c. **b** *Left*, cartoon representation of hCdc34BΔdist-Ub as in **a** but rotated ~90° with CC0651 shown as gray sticks and isopeptide bond indicated. *Right*, zoomed in view of Cdc34 CTDprox, Ubc core, and Ub(t) interfaces with contacting residues shown as sticks. **c** K48-diUb assays for the indicated mutants with (*top*) and without (*middle*) minimal E3 ligase. Data are represented by mean ± SD with individual replicates shown as gray circles. Biochemical assays have four independent replicates, unless indicated by an asterisk when there are three replicates. Representative replicates are labeled above. *Bottom*, single turnover K48-diUb assay with indicated Ub mutants (*n* = 4). Data are shown as above with representative image labeled above with (−) lanes to show initial thioester formation and (+) to indicate subsequent addition of E3 ligase and excess UbA for single Ub discharge. Source data are provided as a Source Data file

specifically, control of cell cycle progression via K48-polyUb-mediated degradation of cyclin-dependent kinase (CDK) inhibitors[65]. To test this, we selected three Cdc34A mutants with diminished activities in vitro (Figs. 5d and 6c), including V183D, Y190D/C191D (YC→DD), and the proxmut construct with the CTDprox replaced by a flexible linker. As expected, knockdown (KD) of Cdc34A in U2OS human osteosarcoma cells using two independent short-hairpin RNAs (shRNAs) led to increased p27 levels[66] (Fig. 7a and Supplementary Fig. 7). Furthermore, overexpression (OE) of WT Cdc34 reversed p27 protein elevation in Cdc34 KD cells, while OE of Cdc34 mutants V183D, YC→DD, and proxmut were not able to do so (Fig. 7b and Supplementary Fig. 7). To study the potential effects of Cdc34 CTDprox disruption on cell cycle progression, U2OS cells were subjected a flow cytometry-based cell cycle assay. First, U2OS cells treated with control or Cdc34 shRNA were synchronized in G2 phase (Fig. 7c, *left*) followed by release for twelve hours (Fig. 7c, *right*). DNA content was tracked by propidium iodide (PI)

incorporation combined with flow cytometric analysis, and gates were indicated for G1 (magenta), S (blue), and G2/M (cyan) phases. Upon release, more cells with Cdc34 KD were retained in G1 phase with an increase from 37 to 45% compared to control cells (Fig. 7c, d). Although these differences were subtle, they were consistent over three independent experiments and were statistically significant (Fig. 7d), suggesting that the observed retention of live cells in G1 is indeed due to Cdc34A KD in the cells that may be partially compensated for by the Cdc34B isoform or even other E2s. Subsequently, U2OS cells with Cdc34 KD plus OE of WT or mutant Cdc34 were subjected to the same flow cytometric analysis. OE of WT Cdc34 was able to rescue the increase of cells in G1 phase from Cdc34 KD; whereas, OE of Cdc34 mutants was unable to rescue this effect, consistent with the earlier in vitro findings (Fig. 7d, e). Interestingly, U2OS cells expressing the Cdc34 mutants V183D, YC→DD, and proxmut displayed a phenotype similar to cells expressing catalytically inactive Cdc34 mutant, C93A (Fig. 7d, e).

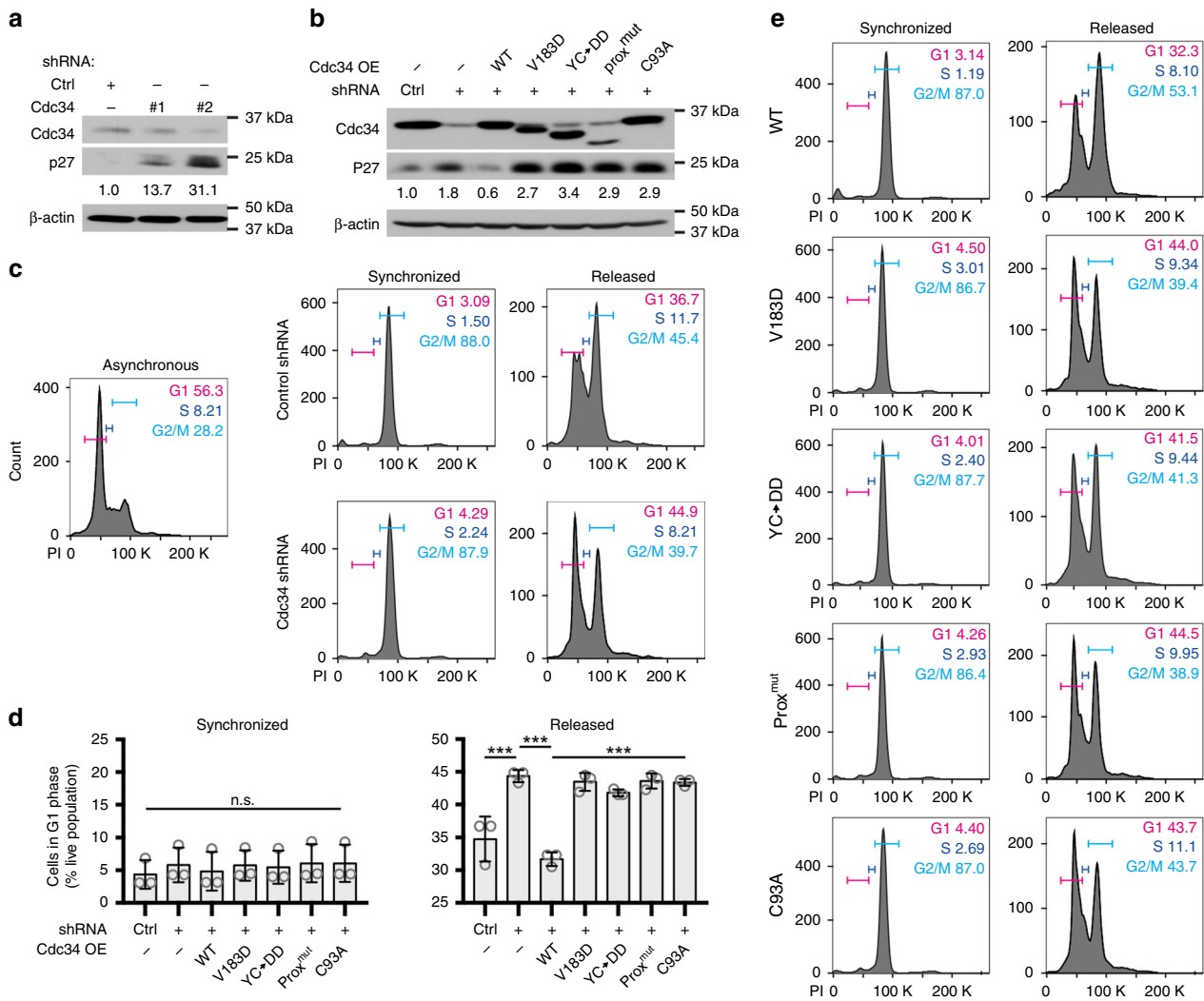

**Fig. 7** Cdc34 CTDprox contacts to Ub(t) and the Ubc core are important for Cdc34 function in cells. **a**, **b** Representative western blots (for $n = 3$) of U2OS cells for Cdc34 and p27 after Cdc34 shRNA KD (**a**) and subsequent overexpression of WT or mutant Cdc34 (**b**). **c** Flow cytometric analysis of U2OS cells treated with control or Cdc34 shRNA then synchronized in G2 phase with nocodazole (*middle*) and released from synchrony for 12 h (*right*) with propidium iodide staining to track DNA content. *Left*, control asynchronous U2OS cell population subjected to the same analysis. Gates indicate percent of live cell population in stages of cell cycle indicated. **d** Graphical representation of three independent cell cycle experiments as in **c** and **e** with the percent of live cells in G1 phase during G2 synchronization (*left*) and after 12-h release (*right*) represented by the mean ± SD with individual replicates shown as gray circles. Statistical analysis was conducted using a one-way ANOVA with Bonferroni as Post Hoc test ($n = 3$), and n.s. indicates non-significant changes while *** indicates a $p$-value of < 0.001. **e** Flow cytometric analysis as in **c** with Cdc34 shRNA KD followed by WT or mutant Cdc34 overexpression as indicated by labels. Source data for all panels are provided as a Source Data file

These findings suggest that the selected Cdc34 mutants are defective in polyubiquitination, leading to the accumulation of p27 and slowed progression of cell cycle. While the structural and biochemical data strongly suggest that this phenotype is due to the disruption of inherent Cdc34 CTDprox function by destabilizing the Ub(t) closed conformation, other potential mechanisms may exist in the cell such as the disruption of Cdc34/SCF binding and further study is required to dissect the detailed molecular mechanisms. Altogether, the results of the structural, biochemical, and cell-based assays support the importance of intra- and intermolecular CTDprox interactions for Cdc34 K48-polyUb chain formation on protein targets in cells and, subsequently, regulation of cell cycle progression.

## Discussion
In this manuscript, we have presented a series of structural snapshots that mimic the substrate complex and product of Uba1-Cdc34 Ub thioester transfer. Our Cdc34apo and Uba1-Cdc34 structures reveal combinatorial Uba1 recognition of Cdc34 involving a distinct Cdc34/UFD interface and conformational changes in Cdc34 that accompany Uba1 binding. Comparative analysis reveals that the Uba1 UFD utilizes a conserved core of residues to build distinct networks of contacts with divergent E2 hA sequences, highlighting UFD plasticity. Further, a flexible Uba1 architecture allows E1 and E2 active sites to come together during thioester transfer despite significant differences i~n E2-binding modes. These findings could provide broader insights into the molecular basis for promiscuity/specificity in protein–protein interactions, particularly with regard to protein adaptation to dynamic environments and how evolution shapes protein interaction networks.

Our Cdc34-Ub structure reveals that elements of the Cdc34 CTDprox become ordered upon transfer of Ub to Cdc34 and engage in contacts to Ub(t) that stabilize an active Cdc34~Ub

closed conformation. This discovery answers a longstanding question regarding the structural basis by which CTD$^{prox}$ promotes Cdc34 activity. Notably, Cdc34/Ub(t) interactions are facilitated by intramolecular interactions between CTD$^{prox}$ and the previously identified E2 backside surface that interacts with non-covalent Ub for allosteric activation in a subset of E2s. Many E2s harbor uncharacterized C-terminal extensions and our studies raise the possibility that these regions may also play a role in E2 activity through contacts to Ub(t) or allosteric activation. Altogether our structural, biochemical, and cell-based studies reveal the molecular underpinnings for two critical Cdc34 activities: engaging Uba1 for E1-E2 thioester transfer and positioning of the subsequent thioester-bound Ub molecule for optimal Ub discharge for Lys48-linked polyUb chain extension. These findings provide a step forward in the progress of understanding Cdc34's cellular functions and lay groundwork for future investigations of Cdc34 and SCF ligase interactions.

## Methods

**Cloning**. _S. cerevisiae_ Uba1(a.a. 11–1024) and Cdc34$^{FL}$ were cloned from genomic DNA into the _NcoI_ and _XhoI_ sites of pET29NTEV. Human Cdc34A$^{FL}$ and Cdc34B$^{FL}$ were amplified from a cDNA library and cloned into the _NcoI_ and _XhoI_ sites of pET29NTEV. Human and _S. pombe_ Uba1 were cloned as previously described[33]. _S. pombe_ ubiquitin was cloned into pET28 with a non-cleavable N-terminal His tag. Human ubiquitin was cloned from cDNA into the _NcoI_ and _XhoI_ sites of pET29NTEV. All point mutations were introduced using PCR-based site-directed mutagenesis. Truncation mutants of Cdc34 were introduced by mutating a stop codon at the desired site. Mutated and deleted CTD$^{prox}$ constructs were synthesized then inserted into the _NcoI_ and _XhoI_ of pET29NTEV. Human Cul1 (a. a. 411–690) and _S. cerevisiae_ Cul1 (a.a. 429–814) were synthesized and inserted in the _BamHI_ and _NotI_ sites of pSMT3. Human Rbx1 (a.a. 5–108) and _S. cerevisiae_ Rbx1 (1–121) were cloned from a synthesized construct and inserted into _NcoI_ and _XhoI_ of pETduet. All constructs and point mutations were generated using the primer pairs described in Supplementary Table 2.

**Protein expression and purification**. All proteins were expressed in _E. coli_ strain BL21 (DE3) Codon Plus (Stratagene). Rbx1 and Cul1 constructs were co-expressed using dual antibiotic selection. Large-scale cultures were grown at 37 °C until the desired A600 OD was reached then placed in an ice bath cold shock with 1.5% ethanol. After 30 min, isopropyl-β-D-1-thiogalactoside (IPTG) was added to a desired concentration of 0.1–0.5 mM, then protein expression was induced at 18 °C shaking for 18 h. At induction, 0.15 mM zinc acetate was added to Cul1/Rbx1 co-expression cultures. All proteins were grown in Luria Broth to an OD of 1.0–2.0 except for _S. cerevisiae_ Uba1, which was grown to saturation in Terrific Broth overnight prior to induction. After induction, cells were pelleted by centrifugation then resuspended in 20 mM Tris pH 8.0, 350 mM NaCl and 20 mM imidazole and snap frozen in liquid nitrogen for storage or processed immediately. Frozen pellets were rapidly thawed then 0.5 mM dithiothreitol (DTT), 100 μg DNase per liter of culture, and 1 mg Lysozyme per liter of culture were added and cells were lysed by sonication. Cell lysis supernatant was cleared by centrifugation then applied to Ni·NTA superflow resin (QIAGEN) by gravity flow. Bound protein was eluted in 20 mM Tris pH 8.0, 350 mM NaCl, 250 mM imidazole, 2 mM DTT. Purified protein was subjected to gel filtration by Superdex 200 or Superdex 75 based on protein size (Amersham), tag cleavage overnight where applicable, anion exchange purification of a MonoQ column (Amersham), then buffer exchange to 20 mM Tris-HCl pH 8.0, 50 mM NaCl. The exception to this protocol were Cdc34 WT and mutants and SpUb mutants used for biochemistry. These proteins were purified using limiting amounts of Ni·NTA resin then passed over a desalting column (Amersham) for buffer exchange to 20 mM Tris-HCl pH 8.0, 50 mM NaCl. After purification, proteins were concentrated to 4–10 mg ml$^{-1}$, aliquoted for single use, and snap frozen in liquid nitrogen.

**E1-E2 cross-linking**. E1-E2 cross-linking was performed according to published methods[32,33]. Cdc34 was incubated in fresh activating buffer (20 mM Tris pH 8.0, 50 mM NaCl, 2.5 mM 2,2'-dipyridyldisulfide, 2.5% DMSO) at 22 °C for 15 min followed by filtration and desalting to removed excess 2,2'-dipyridyldisulfide. Then, Uba1 and activated Cdc34 were mixed and incubated for 15 min at 22 °C. Uba1-Cdc34 crosslink product was purified over non-reducing Superdex 200 and MonoQ columns, concentrated to 11.3 mg ml$^{-1}$ and snap frozen in liquid nitrogen.

**Cdc34-Ub conjugation**. Five micromolar Cdc34 active site Cys to Lys mutant was incubated with 0.5 μM Uba1 and 50 μM Ub overnight at 35 °C in 50 mM Tris-HCl pH 9.5, 50 mM NaCl, 5 mM ATP, 10 mM MgCl$_2$, and 1 mM BME. hCdc34B$^{\Delta dist}$-Ub conjugate was purified over a MonoQ column and concentrated to 17 mg ml$^{-1}$ then snap frozen in liquid nitrogen.

**Crystallization and data collection**. Prior to setting crystallization screens, _S. cerevisiae_ Uba1-Cdc34$^{\Delta dist}$ crosslink (8.7 mg ml$^{-1}$, 58 μM, final) was mixed with 116 μM Ub, 1 mM ATP, and 5 mM MgCl$_2$. Sample was subjected to sparse-matrix screening in Intelli-Plate (Art Robbins Instruments) sitting drop format with 0.2 μl sample and 0.2 μl mother liquor at 18 °C. Crystals initially grew in the condition F7 from Index (Hampton Research). Refinements of crystal conditions in a hanging drop vapor diffusion tray with 0.75 μl of sample and 0.75 μl of mother liquor resulted in large crystals growing in 0.2 M ammonium sulfate, 25% PEG 3,350, 0.1 M Bis-Tris pH 6.5. Crystals were equilibrated in mother liquor with cryoprotectant 20% ethylene glycol, 1 mM ATP, and 5 mM MgCl$_2$ and snap frozen in liquid nitrogen for shipment. Data were collected at Advanced Photon Source (APS, Argonne, Illinois, USA), NE-CAT beamline 24-ID-E.

_S. cerevisiae_ Cdc34$^{\Delta dist}$ (10 mg ml$^{-1}$) was subjected to sitting drop screens as above. Initial crystal hit was in F7 from JCSG Core Suite IV (Hampton Research). Condition refinement in a hanging drop diffusion system produced crystals in 0.06 M zinc acetate, 0.108 M sodium cacodylate, 14.4% PEG 8,000, 20% glycerol. Crystals were snap frozen for shipment. Data were collected at APS SER-CAT beamline 22-ID.

Prior to setting crystallization screens, hCdc34B$^{\Delta dist}$-Ub conjugate (737 μM) was mixed with 1.33-fold excess CC0651 (981 μM), Cdc34 inhibitor, as a crystallization aid. Sitting drops screens were set as above. Crystals initially grew in H4 from JCSG Core Suite I (Hampton Research). Refinement in hanging drop system produced crystals in 0.03 M potassium dihydrogen phosphate, 23% PEG 8000. Crystals were equilibrated in mother liquor with cryoprotectant 20% ethylene glycol and snap frozen for shipment. Data were collected at APS NE-CAT beamline 24-ID-E.

**Structure determination and refinement**. A complete dataset for _S. cerevisiae_ Uba1-Cdc34$^{\Delta dist}$ was collected to 2.07 Å resolution. PHASER was used for molecular replacement to place Uba1/Ub(a) using PDB: 3CMM as the search model. Clear density was visible for E2, so SCULPTOR was used to create a model of ScCdc34$^{\Delta dist}$ using Ube2g2 (PDB: 3H8K), which was placed into the available density. After refinement, clear positive density was visible for the extended C-terminus of Cdc34 so residues 171–181 were manually built into the density. The model was refined to $R/R_{free}$ values of 0.187/0.217 via iterative rounds of refinement and rebuilding using PHENIX[67] and COOT[68] and geometry analysis via Mol-Probity[69]. The final model contains two copies of Uba1-Cdc34/Ub(a) in the asymmetric unit with residues 11–773 and 797–1024 of Uba1 (chains A/D), 1–76 of Ub (chains B/E), and 3–102 and 112–181 of Cdc34$^{\Delta dist}$ (chains C/F) built. The model also includes 909 ordered water molecules, 21 ethylene glycol molecules, 15 sulfate ions, 2 magnesium ions, and 2 ATP molecules. The space group is P2$_1$ with dimensions (Å) $a = 126.4$, $b = 68.5$, and $c = 171.7$.

A complete dataset for _S. cerevisiae_ Cdc34$^{\Delta dist}$ was collected to 1.65 Å resolution. Chain C from the Uba1-Cdc34/Ub(a) model was used in PHASER for molecular replacement. The model was refined to $R/R_{free}$ values of 0.171/0.199 via iterative rounds of refinement and rebuilding using PHENIX and COOT. The final model contains one copy of Cdc34$^{\Delta dist}$ asymmetric unit with residues 3–102 and 113–178 ordered. The model includes 103 ordered water molecules, 4 Zn$^{2+}$ molecules, and 3 acetate molecules. The space group is P2$_1$2$_1$2$_1$ with dimensions (Å) $a = 40.1$, $b = 49.0$, and $c = 103.7$.

A complete dataset for human Cdc34B$^{\Delta dist}$-Ub was collected to 1.50 Å resolution. hCdc34A/Ub (PDB: 4MDK) was used in SCULPTOR to create a model for molecular replacement via PHASER. After refinement, clear positive density was observed for the C-termini of Ub and Cdc34 so they were built into the density manually to include an isopeptide bond between Ub Gly76 and Cdc34 Lys93. The model was refined to $R/R_{free}$ values of 0.175/0.202 via iterative rounds of refinement and rebuilding using PHENIX and COOT. The final model contains one copy of Cdc34$^{\Delta dist}$-Ub in the asymmetric unit with residues 1–76 of Ub (chain E), and 5–192 of Cdc34$^{\Delta dist}$ (chain A) built. The model also includes 319 ordered water molecules, 1 phosphate, and 3 ethylene glycol molecules. The space group is P2$_1$2$_1$2$_1$ with dimensions (Å) $a = 44.6$, $b = 55.7$, and $c = 119.8$. All molecular graphics representations were generated using PYMOL. All electron density maps were constructed in PYMOL from 2Fo-Fc maps contoured at sigma level 1.0.

**E1-E2 thioester transfer assay**. E1-E2 thioester transfer assay was adapted from the previously described protocol[33]. Assays were performed with 20 nM E1 (species-matched to E2), 500 nM E2, 5 μM Ub, and 1 mM ATP in 20 mM HEPES pH 7.5, 50 mM NaCl, 5 mM MgCl$_2$ buffer. Reaction was started by addition of ATP and mixing then quenched after 30 s by addition of 2x UREA sodium dodecyl sulfate polyacrylamide gel electrophoresis (SDS-PAGE) buffer. Owing to inherently low activity, Ubc15 thioester transfer assay was conducted with 50 nM E1 for 60 s. Samples (7.5 μl each) were subjected to SDS-PAGE at 150 V. Gels were then stained with SYPRO Ruby (BioRad) and visualized with a ChemiDoc MP (BioRad). Data quantification was conducted using densitometry in ImageJ software with original unedited images. Densitometry measurements were normalized as a percentage of the control WT assay on the same gel. Data are represented as an average of 3–4 technical replicates with +/−1 standard deviation error bars. Unprocessed images of representative gels for all biochemical assays are provided in the Source Data file.

**Multiple-turnover Lys48-linked diUbiquitination assay**. Multiple-turnover Lys48-linked diUbiquitination assay was adapted from a previously described protocol[57]. Assays were performed with 50 nM E1 (species-matched to E2), 500 nM E2, 5 μM Ub (species-matched), +/− 100 nM minimal E3 fragment (species-matched) and 1 mM ATP in 20 mM HEPES pH 7.5, 50 mM NaCl, 5 mM MgCl$_2$ buffer. Reaction was started by addition of ATP and mixing then quenched after 5, 30, or 60 m as indicated by addition of 2x UREA SDS-PAGE buffer. Samples (7.5 μl each) were subjected to SDS-PAGE, stained, and imaged as for the thioester transfer assays. Data were quantified and represented as for the E1-E2 thioester transfer assay.

**Single turnover Lys48-linked diUbiquitination assay**. Single turnover Lys48-linked diUbiquitination assay was adapted from the previously described protocol[62]. Assays were performed with 500 nM E1, 5 μM E2, 5 μM Ub$^D$, and 5 mM ATP in 20 mM HEPES pH 7.5, 50 mM NaCl, 5 mM MgCl$_2$ buffer. In this assay, human proteins were used except for S. cerevisiae WT Ub and mutants. Reaction was started by addition of ATP and mixing. After 10 m, thioester-charged samples were taken and added to 1/20x diluted Urea SDS-PAGE buffer, and 1 unit apyrase (Sigma) was added. After 5 m, 1 μM minimal E3 fragment and 50 μM Ub$^A$ was added to the reaction with mixing and then quenched after 5 m by addition of 1/20x diluted Urea SDS-PAGE buffer. Samples (0.75 μl each) were subjected to SDS-PAGE, stained, and imaged as for the thioester transfer assays. Data were quantified and represented as for the E1-E2 thioester transfer.

**Cell culture**. U2OS (Cat. # HTB-96) and HEK293T (Cat. # CRL-3216) cells were purchased from the American Type Culture Collection (ATCC). Both cell lines have the authentication information from ATCC, and were tested negative for mycoplasma contamination. HEK293T cells were cultured in Dulbecco's Modified Eagle Medium containing 10% Fetal Bovine Serum (FBS) and 1% penicillin–streptomycin. U2OS cells were maintained in McCoy's 5 A medium with 10% FBS and 1% penicillin–streptomycin. All cells were maintained in a humidified incubator with 5% CO$_2$.

**Transfection, virus production, and infection**. For lentivirus production, lentiviral vectors were co-transfected with pMDLg/pRRE, CMV-VSVG, and RSV-Rev vectors. The pLKO.1 shRNA constructs were purchased from Dharmacon (GE Healthcare Life Sciences): TRC cdc34 shRNAs.

For retrovirus production, pMX-puro empty vector, and vectors with Cdc34 WT, C93A, V183D, Y190D/C191D, and prox$^{mut}$ were co-transfected with either QΨ vector. Virus supernatants were collected 48 and 72 h post transfection. Infections were performed using polybrene. Thereafter, cells were either used for experiments or selected with puromycin for further analyses.

**Western blot analysis**. After treatment, cells were collected and lysed in RIPA buffer (50 mM Tris pH 7.5, 150 mM NaCl, 1.0% NP-40, 0.1% SDS, and 0.5% deoxycholic acid) supplemented with protease and phosphatase inhibitors. Lysates were resolved in 10% SDS-PAGE gels for western blotting. Proteins were transferred to PVDF membrane; after blocking with 5% non-fat milk, membranes were incubated with primary antibodies: Cdc34 (Santa Cruz, sc-28381, 1:1000 dilution), p27 (BD Biosciences, 610242, 1:1000 dilution), p27 (Cell Signaling, 3686, 1:1000 dilution), SKP2 (Santa Cruz, sc-7164, 1:1000 dilution), SKP1 (BD Biosciences, 610530, 1:1000 dilution), and β-actin (Sigma, A5316, 1:50,000 dilution). Thereafter, membranes were washed and incubated with relative secondary antibodies: goat anti-rabbit secondary antibody (Cell Signaling, 7074, 1:5000 dilution), or goat anti-mouse secondary antibody (Cell Signaling, 7076, 1:5000 dilution).

Finally, the signals were visualized by the chemiluminescence system (Perkin Elmer). These experiments were conducted three times. Western blot band quantification was performed using Quantity One (BioRad Laboratories, Inc.) and signals were normalized to the control group. Unprocessed scans of blots are provided in the Source Data file.

**Cell cycle analysis**. U2OS cells with control shRNA or Cdc34 shRNA as well as overexpression of Cdc34 WT or mutants were exposed to nocodazole synchronization for 16 h and then released for 12 h; meanwhile, asynchronized cells were used as a control population. Thereafter, cells were trypsinized and washed with cold PBS. Then, cells were stained with 10 μg ml$^{-1}$ propidium iodide (PI) containing 100 μg ml$^{-1}$ RNaseA before flow cytometry analysis. First, the total number of nuclei events acquired were gated using FSC-A vs. SSC-A plot; then, the singlet nuclei were gated by applying a Propidium Iodide-W vs. Propidium Iodide-A plot; and finally, a histogram of Propidium Iodide-A from the singlet was gated to display the phases of the cell cycle.

**Statistical analysis**. Measurements were taken from distinct samples, and the sample size (n) is indicated in the relative figure legends. Plots were made using GraphPad Prism7. SPSS version 24.0 was used for statistical analyses. One-way ANOVA was applied to compare means. The significance level was set at 0.05 for all analyses.

**Reporting summary**. Further information on research design is available in the Nature Research Reporting Summary linked to this article.

## Data availability

Atomic coordinates and structure factors are deposited in the RCSB with accession codes 6NYA, 6NYD, and 6NYO. The source data underlying Figs. 1a, 2b, c, e, 4b, 5d, 6c, 7a–d, Supplementary Figs. 1b, 5c, 6e, 7a, and 7b are provided as a Source Data file. All other data supporting the findings of this study are available from the corresponding author upon reasonable request.

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

## Acknowledgements

We thank Megan Sheridan for assisting in crystallization of Cdc34apo, and Lingmin Yuan, Christopher Davies, and Miklos Bekes for critically reading the manuscript. X-ray diffraction data were collected at SER-CAT 22-ID and NE-CAT 24-ID-E beamlines at the Advanced Photon Source, Argonne National Laboratory. This work is based upon research conducted at the Northeastern Collaborative Access Team beamlines, which are funded by the National Institute of General Medical Sciences from the National Institutes of Health (P41 GM103403). The Pilatus 6 M detector on 24-ID-C beamline is funded by a NIH-ORIP HEI grant (S10 RR029205). This research used resources of the Advanced Photon Source, a U.S. Department of Energy (DOE) Office of Science User Facility operated for the DOE Office of Science by Argonne National Laboratory under Contract No. DE-AC02-06CH11357. The X-ray crystallography facility used for this work is supported by the Office of the Vice President for Research at the Medical University of South Carolina. The liquid handling robot used was purchased via an NIH Shared Instrumentation Award (S10 RR027139-01). Research reported in this publication was supported by NIH R01 GM115568 and R01 GM128731 (S.K.O.), and NIH P01 CA098101 and R01 CA093237 (J.A.D.). This work was also supported, in part, by NIH F30 CA216921 (K.M.W.), the Abney Foundation (K.M.W.), NIH T32 CA193201 (J.H.A.), and NIH T32 DE017551 (S.Q.). The content of this study is solely the responsibility of the authors and does not necessarily represent the official views of the NIH.

## Author contributions

Structural experiments including crystallization, X-ray data collection/processing, model building/refinement, and structural analyses were performed by K.M.W., J.H.A. and S.K. O. K.M.W. and S.S.A. conducted biochemical assays. S.Q. and J.A.D. designed and conducted mammalian cell assays. The manuscript was written by K.M.W. and S.K.O.

## Additional information

**Competing interests:** The authors declare no competing interests.

