## [Peer Review File · Nature Communications]

Reviewers' comments:

Reviewer #1 (Remarks to the Author):

This outstanding paper probes 2 important functions of the Cdc34 ubiquitin-conjugating enzyme: how it is recognized by Uba1, and how Cdc34 bound donor ubiquitin is activated by Cdc34's conserved acidic C-terminal extension. Regarding the first function, the x-ray structure of a *S. cerevisiae* ternary complex containing Uba1, Cdc34 containing the proximal part of its acidic tail, and ubiquitin-adenylate was determined at high resolution. An extremely thorough analysis of the structure is performed, with insightful comparisons made with other E2-E1 structures. Together, this analysis demonstrates how distinct E2s, that have variable residues both in their helix A regions as well as their N-termini, nevertheless promote binding to the Uba1 UFD domain as well as other Uba1 elements.

Regarding the activation of donor ubiquitin bound to Cdc34, the authors provide compelling biochemical and cell biological evidence for a role of the proximal Cdc34 acidic tail in stabilizing the closed conformation of ubiquitin that has been shown to be optimal for isopeptide bond formation with an additional acceptor ubiquitin.

This paper was a pleasure to read. While the E2-E1 structural analysis will likely be of interest mainly to those working directly in this area, the mechanism of donor ubiquitin activation is quite intriguing, and should be of interest to a general audience. Most of my comments below are minor, with the one exception regarding some of the results in Figure 7. Assuming that these are addressed, I fully endorse publishing this paper in Nature Communications.

Fig 7a-d. While overall the findings from these experiments are compelling, I have some issues. First, Fig 7c doesn't show the extent of Cdc34 over-expression. This is relevant, since in panel b, we can see that Cdc34 is over-expressed compared to endogenous levels. However, notice in panel d that Cdc34 expression levels for WT and mutants are comparable to endogenous.

Regarding interpretation of the results in panel c, the case is made that mutant Cdc34 over-expression competes with endogenous for access to SCF, and their lack of activity explains why poly-ubiquitin chains are not observed on p27 substrate. However, notice that all lanes where mutants are over-expressed show reduced levels of p27 compared to lanes where WT Cdc34 is over-expressed. This is inconsistent with the results in 7b, where WT Cdc34 over-expression reduced p27 levels compared to the mutants.

Another point is that the authors cannot rule out a model where these mutants also affect their binding to SCF, which would also be consistent with their observations. I realize this may seem hard to fathom, say for instance the C93A catalytically inactive Cdc34 mutant, but I've seen cases where mutants like this end up perturbed somehow and do not bind to E3 as efficiently as WT.

I don't think I want to ask for a ton of new experiments to buttress this result. Instead, I'd be okay with the author's acknowledgement in the manuscript that their mutants may be defective in both binding and catalysis, and either of these would result in substrate stabilization in vivo. I also don't feel like panel c is essential for the Figure, and since it does rely on so much over-expression (and also shows the opposite effects on p27 levels as in panel b), I'd recommend removing it as well as panel b. Panel d is by far the strongest, since endogenous Cdc34 levels have been lowered, and ectopic expression of WT or mutant Cdc34 is at levels consistent with endogenous.

The cell cycle analysis in Figure 7e,f shows subtle defects with no estimate of the error of measurement. As it stands, it's not possible to assess the statistical significance of these results.

In terms of necessity of proximal C-terminal tail for di-ubiquitin formation (Fig 5d), it would be informative to make a structurally equivalent mutant with *S. cerevisiae* Cdc34. This is a rather important point since the authors are concluding that the proximal tail has a very important role during catalysis, and showing this for yeast Cdc34 would significantly strengthen their argument.

Minor points

The E1-Ubiquitin-adenylate-E2 structural analysis is extraordinarily complete and is certain to satisfy even the most diligent crystallographer. In my opinion, some of the structural figures that depict detailed intermolecular interactions are bordering on being cluttered (e.g. Figure 4a), and it may be worth considering redoing these figures to not include so much detail to improve clarity.

There are also instances where residues are shown that aren't discussed anywhere in the text. For instance, Ubc15 residue Q8 is shown in Figure 4b, but I couldn't find any discussion of this residue and its role in binding the Uba1 UFD.

I don't feel strongly about these issues, and leave it to the author's discretion regarding any changes they may wish to make to address them.

In Figures 5d,6c: what is meant by +/-? ATP presumably?

Which minimal SCF is used in which assays? The methods note cloning of both a yeast and a human minimal SCF, but it's unclear which is being used in the experiments.

Reviewer #2 (Remarks to the Author):

Williams et al. present a series of structural snapshots tracking activities of the cell cycle E2 enzyme, Cdc34. The first half of the manuscript describes a thorough crystallographic study of *S. cerevisiae* Cdc34 with Uba1-ubiquitin, with a disulfide bond between Uba1 and Cdc34. Mutagenesis corroborates that this mimics the substrate complex. There is strong resemblance to Olsen's structures of *S. pombe* Uba1 with Ubc4 and Ubc15. Comparison to the other E1-E2 structures shows rotation of Ubc4 and Ubc15 relative to UFD, the UFD relative to remainder of E1, and a unique network of Cdc34 contacts with Uba1 especially for the amino terminus. Next, the manuscript turns to the product of the E1 reaction, the Cdc34~ubiquitin intermediate. The authors used mutagenesis and multiple and single turnover ubiquitination assays to dissect the domain contributions. They concluded that the Cdc34 CTD can be subdivided to proximal and distal domains, The data showed, in agreement with published conclusions, that the proximal domain is necessary and sufficient for the inherent ability of Cdc34 to discharge Ub while both the CTDprox and CTDdist are necessary for productive E3 catalysis. To understand the function of CTDprox, the authors solved a crystal structure of human Cdc34B~Ub that was stabilized by CC0651. The structure resembles the published structure of Cdc34~Ub with CC0651 where the CTDprox was not present but also shows how CTDprox stabilizes the closed conformation of the Cdc34B~Ub complex. Mutagenesis studies nicely corroborate the structure and confirm the essentiality of CTDprox in the cell cycle. Overall this is another very high-quality, comprehensive study of E1-E2 and ubiquitinating enzyme-Ub complexes from the Olsen lab. There are many commonalities in the structures of these complexes. The study should be published, but this reviewer is unconvinced that the specific details related to the E2 enzyme of especial importance in cell cycle field are a sufficient advance for the high profile journal Nature Communications.

Specific comments:

1. While the bar graphs are effective at showing lower amounts of ubiquitination by mutants, they could lead the reader to believe the experiments examine kinetics. The mutants could affect V_{max} , K_d , or both. The authors should clarify in the text that the bar graphs are for representation only and that the studies are not of enzyme kinetics.

2. Throughout the paper, the authors make statements that may be technically correct but that downplay the aggregate knowledge of E1-E2, E2~Ub, and Cdc34 complexes. The authors should be careful to appropriately reference established concepts in the field.

Reviewer #3 (Remarks to the Author):

Ubiquitylation is an essential posttranslational modification that is initiated by activation of ubiquitin through E1 enzymes. E1s, of which there is one in yeast but two with ubiquitin-specificity in humans, then need to cooperate with multiple E2 enzymes to shuttle the activated ubiquitin to the specificity-determining E3 ligase. Structural investigation of two E1-E2 complexes from yeast have provided some insight into the mechanism of ubiquitin activation, but many questions remain open. As E1 enzymes are potential targets for therapeutic intervention, understanding their mechanism of action is important for many readers.

In this manuscript, Olsen and colleagues solve a series of structures, including a complex between yeast E1, the E2 enzyme Cdc34, an important cell cycle regulator, and ubiquitin; of apo-Cdc34 that is not bound to E2, and of a ubiquitin-charged Cdc34 that contains previously excluded C-terminal sequences. They performed careful structural and biochemical analyses that merit publication in *Nature Communication* without major revisions.

Most importantly, they underscore the structural flexibility that is required for E1 to allow it to interact with so many E2 enzymes. Cdc34 engages E1 through the same, known, E1 domains as Ubc4 or Ubc15, but there are significant and interesting differences in the relative orientation of E1 domains or E2 helices. Furthermore, they demonstrate structural changes, including partial melting of E2 helices, that are encountered upon binding to the E1. Finally, their structure of charged Cdc34 reveals an important and unknown contribution of residues C-terminal to the UBC domain that stabilize the closed E2-ubiquitin conformation known for RING-dependent ubiquitin transfer. These are important findings that merit immediate publication.

I have only one subtle issue: they mention in the introduction and discussion that Cdc34 might be a therapeutic target. However, they show in their work in cell lines that depletion of Cdc34 or expression of inactive Cdc34 has subtle effects on cell cycle progression in human cells. Together with other data in the field, this indicates that Cdc34 is NOT a good target for drug discovery and they should omit these statements, as they might be misleading to researchers with a less deep understanding of the ubiquitin system or the cell cycle.

Reviewers' comments:

Reviewer #1 (Remarks to the Author):

This outstanding paper probes 2 important functions of the Cdc34 ubiquitin-conjugating enzyme: how it is recognized by Uba1, and how Cdc34 bound donor ubiquitin is activated by Cdc34's conserved acidic C-terminal extension. Regarding the first function, the x-ray structure of a *S. cerevisiae* ternary complex containing Uba1, Cdc34 containing the proximal part of its acidic tail, and ubiquitin-adenylate was determined at high resolution. An extremely thorough analysis of the structure is performed, with insightful comparisons made with other E2-E1 structures. Together, this analysis demonstrates how distinct E2s, that have variable residues both in their helix A regions as well as their N-termini, nevertheless promote binding to the Uba1 UFD domain as well as other Uba1 elements.

Regarding the activation of donor ubiquitin bound to Cdc34, the authors provide compelling biochemical and cell biological evidence for a role of the proximal Cdc34 acidic tail in stabilizing the closed conformation of ubiquitin that has been shown to be optimal for isopeptide bond formation with an additional acceptor ubiquitin.

This paper was a pleasure to read. While the E2-E1 structural analysis will likely be of interest mainly to those working directly in this area, the mechanism of donor ubiquitin activation is quite intriguing, and should be of interest to a general audience. Most of my comments below are minor, with the one exception regarding some of the results in Figure 7. Assuming that these are addressed, I fully endorse publishing this paper in Nature Communications.

We are grateful to the reviewer for their positive comments and thoughtful analysis of our study.

Rebuttal Figure 1. Cell-based ubiquitination analysis of p27 in HEK293T cells.

Cells were transfected with plasmids as indicated in each group. 24 hrs post-transfection, cells were treated with 20 μ M MG-132 for 4 hrs. Then, cells were lysed and subjected to His-pulldown for Ub. Protein samples were resolved by SDS-PAGE gel, and subjected to blotting for indicated proteins. To left of blot images, arrow indicates specific bands; star indicates non-specific bands. SE/LE indicate short or long exposure, respectively.

results of the cell cycle analysis in original Figure 7e,f (now Figure 7c,d,e).

In light of these concerns, and at the recommendation of the reviewer, we have omitted original Figure 7b,c from the revised manuscript.

Regarding interpretation of the results in panel c, the case is made that mutant Cdc34 over-expression competes with endogenous for access to SCF, and their lack of activity explains why poly-ubiquitin chains are not observed on p27

Fig 7a-d. While overall the findings from these experiments are compelling, I have some issues. First, Fig 7c doesn't show the extent of Cdc34 over-expression. This is relevant, since in panel b, we can see that Cdc34 is over-expressed compared to endogenous levels. However, notice in panel d that Cdc34 expression levels for WT and mutants are comparable to endogenous.

We appreciate the reviewer's comments regarding the data in Figure 7 and the resulting conclusions. As outlined below, we have performed additional experiments and edited the figure/text to address these concerns and believe it has led to an improvement in clarity and strengthened our conclusions. We agree with the reviewer's point that the original Figure 7c fails to show the extent of Cdc34 overexpression and have performed a new experiment that includes a negative control that enables an assessment of the levels of Cdc34 variant overexpression compared to endogenous (Rebuttal Figure 1). We apologize for omitting this data from the original manuscript.

We also acknowledge that the levels of Cdc34 overexpression are different in the original Figures 7b and 7d and would like to note that it is difficult to compare the level of Cdc34 overexpression in these experiments for several reasons. First, we utilized two different cell lines in an effort to show cross-applicability of our findings: HEK293T cells (original Figure 7b,c) and U2OS cells (original Figure 7d,e,f). Further, for technical reasons, Cdc34 overexpression in each cell line was achieved through a different method. HEK293T cells were transiently transfected (original Figure 7b,c) while U2OS cells were subjected to initial Cdc34 shRNA KD and then stable retrovirus infection. Due to the shRNA KD and subsequent infection of U2OS cells, we were able to achieve levels of Cdc34 overexpression that are closer to endogenous. We would also like to point out that the p27 levels in U2OS cells (original Figure 7d, now Figure 7b) correlate very well with the

substrate. However, notice that all lanes where mutants are over-expressed show reduced levels of p27 compared to lanes where WT Cdc34 is over-expressed. This is inconsistent with the results in 7b, where WT Cdc34 over-expression reduced p27 levels compared to the mutants.

We failed to note in the Methods or in the Figure 7 legend that the cell-based ubiquitination experiment (original Figure 7c) was conducted in cells treated with MG132 (proteasomal inhibitor). Thus, the lower p27 levels in Cdc34 mutant cells are likely due to technical variance in transfection. Blocking proteasomal degradation allows better examination of polyubiquitinated species but prevents analysis of true stabilization of the target, so original Figure 7c is not comparable to original Figure 7b where target stabilization is being examined in the context of active proteasomal degradation. With this information in mind, the loss of p27 polyubiquitination observed in original Figure 7c is consistent with the increase in p27 levels observed in original Figure 7b. We apologize for this confusion and as noted above have removed original Figures 7b and 7c.

Another point is that the authors cannot rule out a model where these mutants also affect their binding to SCF, which would also be consistent with their observations. I realize this may seem hard to fathom, say for instance the C93A catalytically inactive Cdc34 mutant, but I've seen cases where mutants like this end up perturbed somehow and do not bind to E3 as efficiently as WT.

I don't think I want to ask for a ton of new experiments to buttress this result. Instead, I'd be okay with the author's acknowledgement in the manuscript that their mutants may be defective in both binding and catalysis, and either of these would result in substrate stabilization in vivo. I also don't feel like panel c is essential for the Figure, and since it does rely on so much over-expression (and also shows the opposite effects on p27 levels as in panel b), I'd recommend removing it as well as panel b. Panel d is by far the strongest, since endogenous Cdc34 levels have been lowered, and ectopic expression of WT or mutant Cdc34 is at levels consistent with endogenous.

We fully agree with the reviewer on these points and have added some discussion in the revised manuscript (lines 419-423) to highlight that reduced E3 binding could also potentially account for the observations in our cell-based studies. In addition, we have removed original panels 7b and 7c from this figure following the reviewer's suggestion.

The cell cycle analysis in Figure 7e,f shows subtle defects with no estimate of the error of measurement. As it stands, it's not possible to assess the statistical significance of these results.

This is indeed a good point. We do have 3 replicates for the cell cycle analysis and we have run the statistical analysis and added a new panel to Figure 7 (Figure 7d) to demonstrate the significant, though subtle, difference in G1 cell populations.

In terms of necessity of proximal C-terminal tail for di-ubiquitin formation (Fig 5d), it would be informative to make a structurally equivalent mutant with *S. cerevisiae* Cdc34. This is a rather important point since the authors are concluding that the proximal tail has a very important role during catalysis, and showing this for yeast Cdc34 would significantly strengthen their argument.

Given the intertwined nature of the human/yeast Cdc34 literature, this is a valid point. We have generated the equivalent panel of *S. cerevisiae* Cdc34 mutants and assayed them under the same conditions as the human variants presented in Figure 5d. The yeast Cdc34 variants display a similar phenotype to human in the presence of E3 catalysis (Supplementary Figure 5). Additionally, under E3-free conditions the yeast Cdc34 mutants exhibit a reduction in activity, though the reduction in activity is less severe (Supplementary Figure 5). This data has been added to Supplementary Figure 5 and is now discussed in the paper (lines 313-316).

Minor points

The E1-Ubiquitin-adenylate-E2 structural analysis is extraordinarily complete and is certain to satisfy even the most diligent crystallographer. In my opinion, some of the structural figures that depict detailed intermolecular interactions are bordering on being cluttered (e.g. Figure 4a), and it may be worth considering redoing these figures to not include so much detail to improve clarity.

We appreciate the positive comment and agree some of the figures are a bit cluttered. We have reviewed the detailed panels and have decluttered Figures 2a and 4a to a level that we feel improves clarity without sacrificing information, and have moved the more detailed interaction images from Figure 4a to Supplementary Figure 4 for any curious readers. Thank you for this suggestion.

There are also instances where residues are shown that aren't discussed anywhere in the text. For instance, Ubc15 residue Q8 is shown in Figure 4b, but I couldn't find any discussion of this residue and its role in binding the Uba1 UFD. We agree and have removed such residues from the figures. Thank you.

I don't feel strongly about these issues, and leave it to the author's discretion regarding any changes they may wish to make to address them.

We appreciate the reviewer's suggestions for improving clarity and have incorporated them where we felt it was possible.

In Figures 5d,6c: what is meant by +/-? ATP presumably?

We apologize for the confusion. In Figure 5d +/- does indeed indicate +/- ATP and an ATP label has been added to the main and supplementary figures to clarify this. In the single-turnover experiment presented in Figure 6c, +/- indicates before and after the addition of E3 and acceptor Ub, respectively. This is indicated in the figure legend and, to improve clarity, we have also added an E3/Ub^A label to the panel in the figure.

Which minimal SCF is used in which assays? The methods note cloning of both a yeast and a human minimal SCF, but it's unclear which is being used in the experiments.

We apologize for this omission and have clarified this in the Methods (lines 587, 597-98); like E1, the E3 minimal fragment is species-matched to the E2 being assayed. Now that we have added yeast diubiquitin assays to Supplementary Figure 5 as mentioned above, we have left cloning of both yeast and human E3 in the Methods.

Reviewer #2 (Remarks to the Author):

Williams et al. present a series of structural snapshots tracking activities of the cell cycle E2 enzyme, Cdc34. The first half of the manuscript describes a thorough crystallographic study of S. cerevisiae Cdc34 with Uba1-ubiquitin, with a disulfide bond between Uba1 and Cdc34. Mutagenesis corroborates that this mimics the substrate complex. There is strong resemblance to Olsen's structures of S. pombe Uba1 with Ubc4 and Ubc15. Comparison to the other E1-E2 structures shows rotation of Ubc4 and Ubc15 relative to UFD, the UFD relative to remainder of E1, and a unique network of Cdc34 contacts with Uba1 especially for the amino terminus. Next, the manuscript turns to the product of the E1 reaction, the Cdc34~ubiquitin intermediate. The authors used mutagenesis and multiple and single turnover ubiquitination assays to dissect the domain contributions. They concluded that the Cdc34 CTD can be subdivided to proximal and distal domains. The data showed, in agreement with published conclusions, that the proximal domain is necessary and sufficient for the inherent ability of Cdc34 to discharge Ub while both the CTDprox and CTDdist are necessary for productive E3 catalysis. To understand the function of CTDprox, the authors solved a crystal structure of human Cdc34B~Ub that was stabilized by CC0651. The structure resembles the published structure of Cdc34~Ub with CC0651 where the CTDprox was not present but also shows how CTDprox stabilizes the closed conformation of the Cdc34B~Ub complex. Mutagenesis studies nicely corroborate the structure and confirm the essentiality of CTDprox in the cell cycle. Overall this is another very high-quality, comprehensive study of E1-E2 and ubiquitinating enzyme-Ub complexes from the Olsen lab. There are many commonalities in the structures of these complexes. The study should be published, but this reviewer is unconvinced that the specific details related to the E2 enzyme of especial importance in cell cycle field are a sufficient advance for the high profile journal Nature Communications.

Thank you for the positive commentary on our studies.

Specific comments:

1. While the bar graphs are effective at showing lower amounts of ubiquitination by mutants, they could lead the reader to believe the experiments examine kinetics. The mutants could affect Vmax, Kd, or both. The authors should clarify in the text that the bar graphs are for representation only and that the studies are not of enzyme kinetics.

Thank you for pointing out this potential source of confusion. We have added a clarification in the results when the first bar graph data is presented (lines 110-112).

2. Throughout the paper, the authors make statements that may be technically correct but that downplay the aggregate knowledge of E1-E2, E2~Ub, and Cdc34 complexes. The authors should be careful to appropriately reference established concepts in the field.

We have reviewed the references previously published literature, primarily in the introduction, and found the discussion of E2~Ub/E3 interactions was lacking some important references which are included in the revised manuscript to more clearly highlight established concepts in the field. We worked to incorporate as many references and aggregate knowledge as possible without disregarding the formatting restrictions of the journal.

Reviewer #3 (Remarks to the Author):

Ubiquitylation is an essential posttranslational modification that is initiated by activation of ubiquitin through E1 enzymes. E1s, of which there is one in yeast but two with ubiquitin-specificity in humans, then need to cooperate with multiple E2 enzymes to shuttle the activated ubiquitin to the specificity-determining E3 ligase. Structural investigation of two E1-E2 complexes from yeast have provided some insight into the mechanism of ubiquitin activation, but many questions remain open. As E1 enzymes are potential targets for therapeutic intervention, understanding their mechanism of action is

important for many readers.

In this manuscript, Olsen and colleagues solve a series of structures, including a complex between yeast E1, the E2 enzyme Cdc34, an important cell cycle regulator, and ubiquitin; of apo-Cdc34 that is not bound to E2, and of a ubiquitin-charged Cdc34 that contains previously excluded C-terminal sequences. They performed careful structural and biochemical analyses that merit publication in Nature Communication without major revisions.

Most importantly, they underscore the structural flexibility that is required for E1 to allow it to interact with so many E2 enzymes. Cdc34 engages E1 through the same, known, E1 domains as Ubc4 or Ubc15, but there are significant and interesting differences in the relative orientation of E1 domains or E2 helices. Furthermore, they demonstrate structural changes, including partial melting of E2 helices, that are encountered upon binding to the E1. Finally, their structure of charged Cdc34 reveals an important and unknown contribution of residues C-terminal to the UBC domain that stabilize the closed E2-ubiquitin conformation known for RING-dependent ubiquitin transfer. These are important findings that merit immediate publication.

I have only one subtle issue: they mention in the introduction and discussion that Cdc34 might be a therapeutic target. However, they show in their work in cell lines that depletion of Cdc34 or expression of inactive Cdc34 has subtle effects on cell cycle progression in human cells. Together with other data in the field, this indicates that Cdc34 is NOT a good target for drug discovery and they should omit these statements, as they might be misleading to researchers with a less deep understanding of the ubiquitin system or the cell cycle.

We are grateful to the reviewer for their thoughtful analysis and enthusiasm regarding our work. We have edited the manuscript to remove statements regarding Cdc34's potential as a drug target from the conclusion, and worked to make the point in the introduction a more accurate representation of ongoing work targeting signaling through SCF E3 ligases.

REVIEWERS' COMMENTS:

Reviewer #1 (Remarks to the Author):

The authors have fully addressed all of my concerns, and I endorse publication of the manuscript in Nature Communications.